# Stochastic Process Learning via Operator Flow Matching

## Abstract

Using neural operators, we propose a novel framework for stochastic process learning across arbitrary domains. In particular, we develop operator flow matching for learning stochastic process priors on function spaces. Operator flow matching provides the probability density of any finite collection of points, and enables mathematically tractable functional regression at new points with mean and density estimation. Our method outperforms state of the art models at stochastic process learning, functional regression, and prior learning.

## 1 Introduction

Stochastic processes are foundational to many domains, from functional regression and physics reanalysis, to financial markets, geophysics, and black box optimization. These processes inherently involve stochasticity, can serve as prior distributions over functions, and provide the density of any finite collection of points. Conventionally, priors over processes are hand-designed from predefined Gaussian processes (GP) and their variants tuned against data, only allowing for GP regression. This is despite the fact that phenomena modeled in the natural world often do not follow Gaussianity Fig 1. Consequently, this hinders the flexibility and generalizability of these stochastic processes in real-world applications, leaving behind significant challenges for more general stochastic process learning (SPL).

In SPL, the prior over the stochastic process is learned from data, i.e., historical point evaluation of past experiments. Learning the prior over the process is crucial for universal functional regression(UFR) which is a recently proposed Bayesian method for functional regression and takes GP-regression as its special case when the prior is Gaussian (Shi et al., 2024a). UFR is important to scientific and engineering domains, including reanalysis, data completion, and uncertainty quantification, as well as black box optimization, to name a few.

In this paper, we introduce a new operator learning framework for learning stochastic process priors and performing efficient UFR, based on a generalization of marginal optimal-transport flow matching (Tong et al., 2024), subsequently referred to as operator flow matching (OFM). To achieve this, we extend neural operators (Azizzadenesheli et al., 2024)–designed initially to map functions between infinite-dimensional spaces–to maps between collections of points by leveraging their functional convergence properties. These serve as the main architecture blocks in OFM. We propose to generalize marginal optimal transport flow matching to arbitrary collections of points, allowing us to learn probability priors over the stochastic process, ergo, sampling the value of any collection points with their associated density.

After learning the prior and having access to densities, OFM can be used for UFR, where given any collection of points of the underlying function, we estimate the mean value of any new collection of points and efficiently sample from their posterior values using stochastic gradient Langevin dynamics (SGLD) (Welling & Teh, 2011). We show that OFM outperforms previous state of the art methods, including deep GPs, neural processes, and the state-of-the-art operator flow (OpFlow) (Salimbeni & Deisenroth, 2017; Jankowiak et al., 2020; Garnelo et al., 2018; Kim et al., 2019; Shi et al., 2024a).

In the preliminary section3, we explain the SPL problem setup, define UFR, and generalize the flow matching formulation to stochastic processes. In the method section 4, we propose marginal optimal transport flow matching, generalized to stochastic processes, and we show how this model can be used for SPL, distributional informed UFR, and also as a functional generative model.

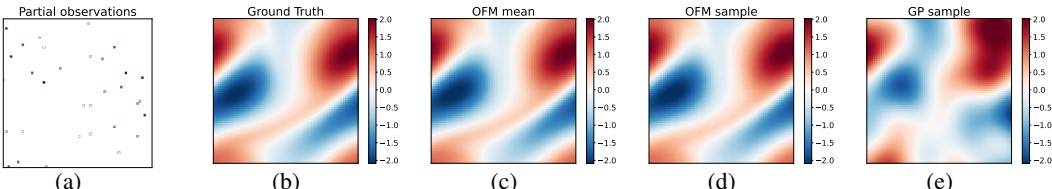

Figure 1: Operator Flow Matching (OFM) regression on Navier-Stokes functional data with resolution $64 \times 64$. (a) 32 random observations. (b) Ground truth sample (c) Predicted mean from OFM. (d) One posterior sample from OFM. (e) One posterior sample from best fitted GP.

To summarize, OFM is the first simulation-free ODE framework that transports a Gaussian process to a target stochastic process for functional regression purpose, enabling likelihood estimation for any collection of points. Compared to existing baselines in functional regression, OFM enjoys greater expressiveness without the model constraints seen in deep GPs or OpFlow, and avoids the theoretical limitations associated with neural processes (see Appendix A.8). We empirically show that regression with OFM outperforms existing baselines, matches classical GP Regression on GP examples, and delivers exceptional performance on highly non-Gaussian functional datasets, such as those from Navier-Stokes equations and black hole simulations.

## 2 RELATED WORK

**Neural operators.** Neural operators constitute a new paradigm in machine learning for learning maps between function spaces, a generalization of conventional neural networks that map between finite dimensional spaces (Li et al., 2021; Kovachki et al., 2023). Among neural operator architectures, Fourier neural operators (FNO) (Li et al., 2021) enable convolution in the spectral domain and have been shown effective in operator learning (Pathak et al., 2022; Wen et al., 2023; Yang et al., 2021; 2023; Sun et al., 2023; Li et al., 2023). In this work, we use this as our choice of neural operators architecture.

**Direct function samples.** There is a body of work on generative models dedicated to learning distributions over functions, such that direct sampling on the function space is possible. For example, generative adversarial neural operators (GANO) generalize generative adversarial nets on finite dimensional spaces to function spaces (Rahman et al., 2022; Shi et al., 2024b), yielding a neural operator generative model that maps Gaussian random fields (GRF) to data functions (Azizzadenesheli et al., 2024). Other works in this area have followed the success of diffusion models (Song et al., 2021; Ho et al., 2020) in finite dimensional spaces, e.g., denoising diffusion operators generalize diffusion models to function spaces by using GRF as a mean of noisification and use neural operators to learn the score operator on function valued data (Lim et al., 2023; Pidstrigach et al., 2023; Kerrigan et al., 2023a). Moreover, the same principle has been deployed to generalize flow matching (Lipman et al., 2023) to functional spaces (Kerrigan et al., 2023b), an approach closely related to our work. However, these works on learning generative models on function spaces do not support UFR the way GP-regression does because they (i) focus solely on generating function samples, (ii) do not clarify how to model a stochastic process on point value sequential generation, and (iii) do not provide point evaluation of probability density.

**Stochastic processes.** Earlier works on SPL have focused on hand-tuned methods in the style of GP-regression. In these cases, an expert tunes the GP parameters given a set of experimental samples. More advanced methods rely on deep GPs, in which a network of GPs is stacked on top of each other. The parameters of deep GPs are commonly optimized by minimizing the variational free energy, which serves as a bound on the negative log marginal likelihood. (Damianou & Lawrence, 2013; Liu et al., 2020). Deep GPs have limitations in terms of learnability, expressivity, and computational complexity. Warped GPs (Kou et al., 2013) and transforming GP (Maroñas et al., 2021) methods use historical data to learn a pointwise transformation of GP values and achieve on par performance compared to deep GP type methods. The pointwise nature of such approaches limits their generality.

Another attempt to address limitations in SPL is neural processes (Garnelo et al., 2018), inspired by variational inference method and designed for sampling from function spaces. This method trains a model to map any collection of points and their values to a vector, used as an input to a decoder that

maps any collection of points to their values. The architectures used in these modes are not consistent as the number of points grows, and same with the decoder, making the approach limited to finite dimensions. The diffusion based variants (Dutordoir et al., 2023) also use uncorrelated Gaussian noise, and the results do not exist in function spaces (Rahman et al., 2022; Lim et al., 2023). In the end, methods based on neural processes still are unable to provide density estimation for collections of points, as needed for UFR.

Finally, OPFLOW introduced invertible neural operators that are trained to map any collection of points sampled from a GP to a new collection of points in the data space (Shi et al., 2024a), using the maximum likelihood principle. This method is consistent as the resolution grows, captures likelihood of any collection of points, and allows for UFR using SGLD. However, similar to normalizing flow (Papamakarios et al., 2021) methods in finite dimensional domains, the use of invertible deep learning models makes their training a challenge, particularly with regards to expressiveness.

## 3 PRELIMINARY

### 3.1 STOCHASTIC PROCESS LEARNING

Let $(\Omega, \mathcal{F}, P)$ denote a probability space and let $(\mathbb{R}^d, \mathcal{B}(\mathbb{R}^d))$ denote a measurable space where $\mathcal{B}(\mathbb{R})$ is the Borel space. Following the standard definition of stochastic processes, a stochastic process $\mathcal{P}$ on a domain $D$ is a collection of $\mathbb{R}^d$-valued random variables indexed by members of $D$, i.e.,

$$\{a(x) : x \in D\}$$

jointly following the probability law $P$. In the special case of Gaussian processes, e.g., Wiener process, following the Gaussian law for $P$, for any collection points $\{x_1, x_2, \ldots, x_n\}$, the random variables $\{a(x_1), a(x_2), \ldots, a(x_n)\}$ are jointly Gaussian, resulting in a function $a$ to be drawn from a Gaussian random field (GRF). Once again, we need to emphasize, $\{a(x_1), a(x_2), \ldots, a(x_n)\}$ is not a dataset but a discretized observation of one function $a$. In practice, the joint probability distribution of the collection of the random variables is unknown a priori, and needs to be learned.

In SPL, we aim to learn an operator $\mathcal{G}$ that maps a base stochastic process $\mathcal{P}$ to another stochastic process $\mathcal{Q}$ that represents the data. That is, for any collection of points $\{x_1, x_2, \ldots, x_n\}$, and for any $n$, the operator $\mathcal{G}$ maps the law on $\{a(x_1), a(x_2), \ldots, a(x_n)\}$ to $\{u(x_1), u(x_2), \ldots, u(x_n)\}$, where $u(x)$ is a pointwise evaluation of function data sample, i.e.,

$$\{u(x_1), u(x_2), \ldots, u(x_n)\} = \mathcal{G}\left(\{a(x_1), a(x_2), \ldots, a(x_n)\}\right).$$

It is convenient to use a GP as the base stochastic process $\mathcal{P}$ for mathematical tractability, i.e.,

$$\{a(x_1), a(x_2), \ldots, a(x_n)\} \sim \mathcal{N}\left(\mathbf{0}, K\left(\{x_1, x_2, \ldots, x_n\}\right)\right)$$

where $K\left(\{x_1, x_2, \ldots, x_n\}\right)$ is a $n \times n$ covariance matrix with entries described by kernel function $k(x_i, x_j)$. Then, the probability of $\{u(x_1), u(x_2), \ldots, u(x_n)\}$, at evaluation points $\{x_1, x_2, \ldots, x_n\}$, for any $n$ and collection of points on $D$ is given by,

$$\mathbb{P}\left(\{u(x_1), u(x_2), \ldots, u(x_n)\}\right) = \mathbf{J}\mathcal{G}\Big|_{\{a(x_1), a(x_2), \ldots, a(x_n)\}} \mathbb{P}\left(\{a(x_1), a(x_2), \ldots, a(x_n)\}\right).$$

where with abuse of notation $\mathbb{P}(u(x))$ denotes the density of $u(x)$ at point $x$, same for $\mathbb{P}(a(x))$, and similarly $\mathbf{J}\mathcal{G}\Big|_{\{a(x_1), a(x_2), \ldots, a(x_n)\}}$ is the Jacobian of the map from the collection of random variables $\{a(x_1), a(x_2), \ldots, a(x_n)\}$ at points $\{x_1, x_2, \ldots, x_n\}$ to random variables $\{u(x_1), u(x_2), \ldots, u(x_n)\}$. It's trivial to verify that $\mathcal{Q}$ is indeed a valid stochastic process via Kolmogorov Extension Theorem (KET) (Kolmogorov & Barucha-Reid, 2018) with a proof provided in Appendix. A.2 . In SPL, we aim to learn a neural operator $\mathcal{G}_\theta$ such that the resulting $\mathcal{Q}$ matches the data process under the true $\mathcal{G}$.

### 3.2 UNIVERSAL FUNCTIONAL REGRESSION

UFR is concerned with Bayesian regression on function spaces (Shi et al., 2024a), where it can be used to infer the posterior of an unknown function on a domain $D$ from a collection of pointwise observations. The observations are often corrupted with noise of variance $\sigma^2$, denoted as

$\{\widehat{u}(x_1), \widehat{u}(x_2), \ldots, \widehat{u}(x_n)\}$ or $\{\widehat{u}(x_i)\}_{i=1}^n$. More specifically, for $m \geq n$ points at which the function is to be inferred,

$$\mathbb{P}\left(\{u(x_1), u(x_2), \ldots, u(x_m)\}\middle|\{\widehat{u}(x_1), \widehat{u}(x_2), \ldots, \widehat{u}(x_n)\}\right)$$

Note that when the prior over the function space is Gaussian, UFR reduces to the celebrated GP regression. Following Bayes rule, and maps between stochastic processes, we obtain the log posterior as follows,

$$\log \mathbb{P}\left(\{u(x_i)\}_{i=1}^m \middle| \{\widehat{u}(x_i)\}_{i=1}^n\right) = -\frac{1}{2}\sum_i^n \frac{(\widehat{u}(x_i) - u(x_i))^2}{\sigma^2} - n\log(\sigma) - \frac{n}{2}\log(2\pi)$$
$$+ \log \mathbb{P}\left(\{u(x_i)\}_{i=1}^m\right) - \log \mathbb{P}\left(\{\widehat{u}(x_i)\}_{i=1}^n\right)$$

This equality holds for any collection of points. It is worth noting that the posterior is exact up to constants, i.e., the second, third, and last terms are constant. Therefore, they do not contribute in MAP estimation, mean estimation, and functional regression in general, and there is no need to compute them.

### 3.3 GENERALIZING FLOW MATCHING TO STOCHASTIC PROCESSES

For any $n$, and points $\{x_1, x_2, \ldots, x_n\}$, consider an ODE system in which a vector of random variables $u_0 \in \mathbb{R}^n$ is gradually transformed into $u_1$, for which, the $i$th entry is equal to $u(x_i)$, via a smooth, time-varying vector field, denoted by $\mathcal{G}_t$,

$$u_t := \phi_t(u_0) = u_0 + \int_0^t \mathcal{G}_s(u_s)ds. \tag{1}$$

Given the density of $p_0 := \mathbb{P}\left(\{a(x_1), a(x_2), \ldots, a(x_n)\}\right)$ where $u_0 \sim p_0$, the time-varying density $p_t$ induced by the diffeomorphism $\phi_t$ or $\mathcal{G}_t$ can be computed using the well-known transport equation (Lipman et al., 2023; Fjelde et al., 2024),

$$\frac{\partial p_t(u_t)}{\partial t} = -(\nabla \cdot (\mathcal{G}_t p_t))(u_t) \tag{2}$$

Eq. 2 shows that constructing $p_t$ is equivalent to constructing $\mathcal{G}_t$ for finite dimensional spaces for which the analysis carries to finite collection of random variables. In the following, we refer to $p_t$ as the marginal probability path induced by $\mathcal{G}_t$ for the given collection of points. From Eq. 2, the log density can be computed through integration,

$$\log p_t(u_t) = \log p_0(u_0) - \int_0^t (\nabla \cdot \mathcal{G}_s)(u_s)ds. \tag{3}$$

In this formulation, we are seeking a specific vector field that transports density $q_0$ to target density $q_1$ for any $n$ and any collection of points $\{x_1, x_2, \ldots, x_n\}$ with boundary conditions $p_0 = q_0, p_1 = q_1$. We propose to extend flow matching (Lipman et al., 2023) to stochastic processes and parameterize a potential vector field $\mathcal{G}_t$ with a neural operator $\mathcal{G}_\theta$, which can be optimized through the flow matching objective for SPL,

$$\mathcal{L}_{\text{FM}}(\theta) := \sup_n \sup_{\{x_1, x_2, \ldots, x_n\}} \mathbb{E}_{t \sim \mathcal{U}(0,1), u_t \sim p_t} \|\mathcal{G}_\theta(t, u_t) - \mathcal{G}_t(u_t)\|^2 \tag{4}$$

Note that $p_t$ and $u_t$ depend on the collocation points. In the above equation, the suprema are intractable and we replace them with expectation as a soft approximation. Moreover, the true $\mathcal{G}_t$ is usually unknown and to address it, one can derive a probability path conditioned on latent variable $z$ of the same alphabet size as the collection. Consequently, the marginal probability path $p_t(u_t)$ is a mixture of conditional probability paths $p_t(u_t|z)$,

$$p_t(u_t) = \int p_t(u_t|z)q(z)dz \tag{5}$$

$$\mathcal{G}_t(u_t) = \mathbb{E}_{q(z)}\left[\frac{\mathcal{G}_t(u_t|z)p_t(u_t|z)}{p_t(u_t)}\right]. \tag{6}$$

Given Eq. 6, the conditional flow matching (CFM) objective is defined as

$$\mathcal{L}_{\text{CFM}}(\theta) := \mathbb{E}_n \mathbb{E}_{x_1, x_2, \ldots, x_n} \mathbb{E}_{t, q(z), p_t(u_t|z)} \| \mathcal{G}_\theta(t, u_t) - \mathcal{G}_t(u_t|z) \|^2 \tag{7}$$

Equations 4, when suprema are replaced with expectations, and 7 have identical gradient for $\theta$, which indicates $\nabla_\theta \mathcal{L}_{\text{FM}}(\theta) = \nabla_\theta \mathcal{L}_{\text{CFM}}(\theta)$. In Flow Matching, the variable $z$ is chosen as a single data point $u_1 \sim q_1$. Considering the class of Gaussian conditional probability paths $p_t(u_t|u_1) = \mathcal{N}(u_t|\mu_t(u_1), \sigma_t(u_1)^2 K(\{x_1, x_2, \ldots, x_n\}))$, with conditional flow $\phi_t(u_t|u_1) = \sigma_t u_0 + \mu_t$. Specially, we choose $\mu_t = tu_1$ and $\sigma_t = 1 - (1 - \sigma)t$, where $\sigma > 0$ is a small constant. Then we can derive a closed-form expression for both the conditional probability and corresponding vectorfield (Tong et al., 2024). Detailed derivation provided in Appendix A.1

$$p_t(u_t|u_1) = \mathcal{N}(u_t|tu_1, (t\sigma - t + 1)^2 K(\{x_1, x_2, \ldots, x_n\})) \tag{8}$$

$$\mathcal{G}_t(u_t|u_1) = \frac{u_1 - (1 - \sigma)u_t}{1 - (1 - \sigma)t} \tag{9}$$

While the conditional vector field $\mathcal{G}_t(u_t|u_1)$ induces an optimal transport path from $p_0(u_t|u_1)$ to $p_1(u_t|u_1)$, the induced marginal path $p_t(u_t)$ is curved and not an optimal-transport path from prior distribution $q_0(u_t)$ to the data distribution $q_1(u_t)$ in general. To address this, Tong et al. (2024) introduced marginal optimal-transport flow matching in finite-dimensional spaces, which takes the conditional variable $z$ from a joint distribution $\pi(u_0, u_1)$ combined with minibatch optimal transport to approximate true marginal (or dynamic) optimal transport. This marginal optimal transport path is a simpler trajectory, resulting in faster training and inference, as well as higher-quality samples compared to the path defined in flow matching approach. In this work, establishing the above formulation, we extend the above-developed flow matching formulation on the stochastic process to their marginal optimal transport one as well as to SPL.

## 4 METHODS

In this section, we first introduce the framework of OFM, which extends marginal optimal transport flow matching (Tong et al., 2024) to infinite-dimensional function spaces. Then, we show how to model a stochastic process and efficiently evaluate exact and tractable likelihoods for any point evaluation of functions with OFM. Lastly, we demonstrate how to use OFM for the UFR setting.

### 4.1 FRAMEWORK OF OPERATOR FLOW MATCHING

For a real separable Hilbert space $(\mathcal{H}, \langle \cdot, \cdot \rangle, \|\cdot\|)$, equipped with the Borel $\sigma-$ algebra of measurable sets denoted by $\mathcal{B}(\mathcal{H})$, we introduce two measures on $\mathcal{B}(\mathcal{H})$: $\nu_0$ as the reference measure and $\nu_1$ as the data measure. Consider a function $h_0$ sampled from $\nu_0$, such that $h_0 \sim \nu_0$. A smooth time-varying functional vector field $\mathcal{G}_t : \mathcal{H} \to \mathcal{H}$ then defines an ordinary differential equation

$$\frac{\partial \phi_t(h)}{\partial t} = \mathcal{G}_t(\phi_t(h)), \tag{10}$$

with initial condition $\phi_0(h_0) = h_0$, where $\phi_t(h)$ the solution of Eq. 10, and $t \in [0, 1]$. Thus, $\phi_t(h)$ represents a function $h$ transported along a vector field from time 0 to time $t$. The diffeomorphism $\phi_t$ induces a pushforward measure $\mu_t := [\phi_t]_\sharp(\mu_0)$, with $\mu_0 = \nu_0$, and we refer to $\mu_t$ as the path of probability measure. The goal is to construct a path of probability measure such that at $t = 1$, $\mu_1 \approx \nu_1$. The dynamic relationship between the time varying measure $\mu_t$ and vector field $\mathcal{G}_t$ can be characterized by the continuity equation:

$$\frac{\partial \mu_t}{\partial t} = -\nabla \cdot (\mu_t \mathcal{G}_t) \tag{11}$$

In practice, we use Eq. 11 in its weak form (Ambrosio et al., 2008; Kerrigan et al., 2023b) to check whether a given vector field $\mathcal{G}_t$ generates the target $\mu_t$:

$$\int_0^1 \int_\mathcal{H} \frac{\partial \varphi(g, t)}{\partial t} + \langle \mathcal{G}_t(g), \nabla_g \varphi(g, t) \rangle d\mu_t(g) dt = 0, \quad \forall \varphi \in \mathcal{C}_c^\infty(\mathcal{H} \times [0, 1]) \tag{12}$$

Suppose that the time-varying vector field $\mathcal{G}_t$ and induced $\nu_t$, which satisfy Eq.12, are known. We can parameterize $\mathcal{G}_t$ with neural operator $\mathcal{G}_\theta : [0, 1] \times \mathcal{H} \to \mathcal{H}$. We can regress $\mathcal{G}_\theta$ to target $\mathcal{G}_t$ through flow matching objective.

$$\mathcal{L}_{\text{FM}}^\dagger = \mathbb{E}_{t \sim \mathcal{U}[0,1], g \sim \mu_t} \| \mathcal{G}_\theta(t, g) - \mathcal{G}_t(g) \|^2 \tag{13}$$

However, similar to its finite-dimensional counterpart, $\mathcal{G}_t$ is typically unknown. Moreover, there are infinitely many paths of probability measures that satisfy the Eq. 12 and ensure $\mu_1 \approx \nu_1$. Therefore, it is necessary to specify a path of probability measures to effectively guide the learning of $\mathcal{G}_\theta$.

## 4.2 CONDITIONAL PROBABILITY MEASURES AND GAUSSIAN MEASURES

Consider a joint probability measure $\pi(\nu_0, \nu_1)$ on $\mathcal{H} \times \mathcal{H}$, where the reference measure $\nu_0$, is chosen as a Gaussian measure, whose absolute continuity is well-studied (Bogachev, 1998). We characterize $\nu_0$ by a Gaussian process with trace-class covariance operator. e.g. $\nu_0 = \mathcal{N}(m_0, C_0)$, where $m_0$ is the mean, $C_0$ is the covariance operator. With the joint measure $\pi(\nu_0, \nu_1)$, we sample a function pair $z := (h_0, h_1)$.

Assuming $\nu_1$ has full support on the Cameron-Martin space associated with $\nu_0$, we construct a conditional probability measure $\mu_t(\cdot|z)$ as a Gaussian measure with trace-class covariance operator and small operator norm to approximate Dirac measures in the sense of weak convergence. Such that, at $t = 0$ and $t = 1$, $\mu_t(\cdot|z)$ is a centered around $h_0, h_1$, approximating $\delta_{h_0}, \delta_{h_1}$ respectively; Subsequently, we can construct a new marginal probability measure by mixing these approximated Dirac measures:

$$\mu_t(A) = \int \mu_t(A|z) d\pi(z), \ \forall A \in \mathcal{B}(\mathcal{H}) \tag{14}$$

Due to $d\pi(z)$ being always positive, the conditional probability measure (Dirac measure approximated by Gaussian measure) is absolutely continuous with respect to $\mu_t$. Eq. 14 indicates that $\mu_0 = \int \delta_{h_0} d\pi(z) \approx \nu_0$, and $\mu_1 = \int \delta_{h_1} d\pi(z) \approx \nu_1$. This formulation suggests that $\mu_0, \mu_1$ represent convolutions of $\nu_0, \nu_1$ with Gaussian measures. For a more detailed discussion on convolution with Gaussian measures, we refer the readers to Appendix B.1 of (Lim et al., 2023).

Suppose $\int_0^1 \int_{\mathcal{H}} \int_{\mathcal{H} \times \mathcal{H}} \|\mathcal{G}_t(g|z)\| d\mu_t(g|z) d\pi(z)$ is finite to guarantee the vector field is sufficiently regular. Under this condition, the vector field that generates $\mu_t$ as specified in Eq. 14 and Eq. 12 can be expanded as follows :

$$\mathcal{G}_t(g) = \int_{\mathcal{H} \times \mathcal{H}} \mathcal{G}_t(g|z) \frac{d\mu_t(\cdot|z)}{d\mu_t}(g) d\pi(z) \tag{15}$$

Eq. 15 is a straightforward extension of the Theorem 1 as detailed in Kerrigan et al. (2023b), we direct readers to Appendix A.1 of Kerrigan et al. (2023b) for more details. We note that $\mu_t(\cdot|z)$ is a Gaussian measure and can be expressed as $\mu_t(\cdot|z) = \mathcal{N}(m_t, C_t)$, with mean $m_t$ and trace-class covariance operator $C_t$. Inspired by Tong et al. (2024), we choose $m_t$ and $C_t$ to have the following forms:

$$m_t = t \cdot h_1 + (1 - t) \cdot h_0 \tag{16}$$

$$C_t = \sigma_{\min}^2 C_0 \tag{17}$$

where $C_0$ is the same Gaussian covariance operator defined for $\mu_0$ and $\sigma_{\min}$ is a small constant. Further, similar to finite-dimensional flow matching, we only consider the simplest vector field that applies a canonical transformation for Gaussian measures, such that the flow has the form: $\phi_t(h|z) = m_t + \sigma_{\min} h_0 \approx t \cdot h_1 + (1 - t) \cdot h_0$. From Eq. 10, we can get $\mathcal{G}_t(h|z) = h_1 - h_0$, indicating $\mathcal{G}_t(h|z)$ is independent of the time $t$ and the path from $h_0$ to $h_1$ is a direct, straight line. Equipped with well-constructed conditional vector field and probability measures, we can train a neural operator $v_\theta$ with the conditional flow matching loss

$$\mathcal{L}_{\text{CFM}}^\dagger = \mathbb{E}_{t \sim \mathcal{U}[0,1], g \sim \mu_t, z \sim \pi(\nu_0, \nu_1)} \|\mathcal{G}_\theta(t, g) - \mathcal{G}_t(g|z)\|^2. \tag{18}$$

Next, we explore how to approximate the true optimal transport plan from optimal coupling of the joint measure $\pi(\nu_0, \nu_1)$. A common way for measuring the distance between two probability measure is 2-Wasserstein distance, which a special case of static Kantorovich formulation (Kantorovich & Rubinshtein, 1958). The static 2-Wasserstein distance is defined as follows

$$W_{\text{sta}}(\nu_0, \nu_1)_2^2 = \inf_{\pi \in \Pi} \int_{\mathcal{H} \times \mathcal{H}} \|h_0 - h_1\|^2 d\pi(h_0, h_1) \tag{19}$$

In the ODE framework, we also care about the dynamic form of the 2-Wasserstein distance to estimate the cost along the transport trajectory, which also is a special case of dynamic Kantorovich

formulation (Chizat et al., 2018).

$$W_{\text{dyn}}(\nu_0, \nu_1)_2^2 = \inf_{\mu_t, \mathcal{G}_t} \int_{\mathcal{H}} \int_0^1 \|\mathcal{G}_t(g)\|^2 d\mu_t(g) dt \tag{20}$$

As stated in Step 3 of Proof of Theorem 4.3 of Chizat et al. (2018), for general measures $\nu_0, \nu_1$, we have $W_{\text{sta}} \leq W_{\text{dyn}}$. However, within the OFM framework, the marginal probability measure is a sum of Dirac measures as described in Eq. 14, and we selected $\nu_0$ as a Gaussian measure and assumed $\nu_1$ has full support on the Cameron-Martin space associated with $\nu_0$. Furthermore, the cost function of 2-Wasserstein distance is squared $L^2$ norm, which is continuous by nature. According to Lemma 4.4 of (Chizat et al., 2018), $W_{\text{sta}} = W_{\text{dyn}}$ for our specifically constructed $\mu_t$ and $\mathcal{G}_t$ in the sense of weak convergence. Therefore, to get the dynamic optimal transport plan, we only need to find a joint measure $\pi(\nu_0, \nu_1)$ that achieves the infimum in Eq. 19. In practice, we use a minibatch approximation of optimal coupling between $\nu_0$ and $\nu_1$. The above approach extends the dynamic (marginal) optimal transport framework of (Tong et al., 2024) to infinite-dimensional function space.

### 4.3 Likelihood estimation and Bayesian Universal Functional Regression

We parameterize $\mathcal{G}_\theta$ with FNO (Li et al., 2021) to ensure our model is resolution agnostic, and assume $\mathcal{G}_\theta$ learns the map from $\nu_0$ to $\nu_1$, which serves as the prior. In practice, we deal with discretized evaluations of functions that may have different sampling rate and resolution. For instance, consider a function $u$ sampled from $\mu_1$, observed on a collection of points $f_1 := \{u(x_1), u(x_2), ..., u(x_m)\}$; thus we have a density function $\mathbb{P}(f_1)$ defined on collection of points $\{x_1, x_2, ..., x_m\}$, where $\mathbb{P}(f_1)$ is derived from measure $\mu_1$. This is similar to how a multivariate Gaussian distribution can be derived from a Gaussian measure characterized by a Gaussian process. Therefore, we can rewrite Eq. 3 as:

$$\log \mathbb{P}(f_1) = \log \mathbb{P}(f_0) - \int_0^1 (\nabla \cdot \mathcal{G}_\theta)(f_t) dt, \tag{21}$$

where $f_0$ and $f_t$ are drawn from the reference Gaussian measure $\nu_0$ and $\nu_t$, respectively, which are also defined on the collection of point $\{x_1, x_2, ..., x_m\}$. Thus $\mathbb{P}(f_0)$ is a multivariate Gaussian with a tractable density function. Furthermore, with the probability density function $\mathbb{P}(f_1)$, we can evaluate the precise likelihood of any $f_1$ from $\mathbb{P}(f_1)$ via Eq. 21. However, following a similar argument to Grathwohl et al. (2018), the computation of $\nabla \cdot \mathcal{G}_\theta(f)$ incurs a cost of $\mathcal{O}(D^2)$ where $D$ is cardinality of set $\{x_1, x_2, ..., x_m\}$. This quadratic time complexity renders the likelihood calculation prohibitively expensive. To address this issue, we adopt the strategy proposed in Grathwohl et al. (2018), utilizing the unbiased Skilling-Hutchinson trace estimator (Hutchinson, 1989; Skilling, 1989) to approximate the divergence term. This technique reduces the computation cost to $\mathcal{O}(D)$, which is the same as the cost of inference, thereby streamlining the evaluation process. The estimator is implemented as follows:

$$\nabla \cdot \mathcal{G}_\theta(f) = \mathbb{E}_{p(\varepsilon)}[\varepsilon^T \frac{\partial \mathcal{G}_\theta(f, t)}{\partial f} \varepsilon] \tag{22}$$

In the unbiased trace estimator, the random variable $\varepsilon$ is characterized by $\mathbb{E}(\varepsilon) = 0$ and $\text{Cov}(\varepsilon) = I$. The gradient computation in Eq. 22 can be efficiently handled with reverse-mode automatic differentiation, allowing for precise estimation with arbitrary error by averaging over a sufficient number of runs, which can benefit from parallel computing of GPUs.

With the efficient tool established for estimating the likelihood of any discretized function samples, we now turn our attention to Bayesian functional regression. Consider a collection of pointwise observations of the underlying unknown function drawn from $\mu_1$, that is corrupted with Gaussian noise, denoted as $\{\widehat{u}(x_1), \widehat{u}(x_2), ..., \widehat{u}(x_n)\}$ or $\{\widehat{u}(x_i)\}_{i=1}^n$. We specifically focus on Gaussian white noise characterized by $\epsilon \sim \mathcal{N}(0, \sigma^2)$, such that $\widehat{u}(x_i) = u(x_i) + \epsilon_i$ for $i \in \{1, \cdots, n\}$. In UFR setting, we are interested in the posterior distribution on new $m \geq n$ points that include the $n$ observation points. With Bayes rule, we have the posterior:

$$\mathbb{P}\left(\{u(x_i)\}_{i=1}^m \Big| \{\widehat{u}(x_i)\}_{i=1}^n\right) = \frac{\mathbb{P}\left(\{\widehat{u}(x_i)\}_{i=1}^n \Big| \{u(x_i)\}_{i=1}^m\right) \cdot \mathbb{P}\left(\{u(x_i)\}_{i=1}^m\right)}{\mathbb{P}\left(\{\widehat{u}(x_i)\}_{i=1}^n\right)} \tag{23}$$

Taking the logarithm of Eq. 23, we have:

$$\log \mathbb{P}\left(\{u(x_i)\}_{i=1}^m \Big| \{\widehat{u}(x_i)\}_{i=1}^n\right) = \log \mathbb{P}\left(\{\widehat{u}(x_i)\}_{i=1}^n \Big| \{u(x_i)\}_{i=1}^m\right) + \log \mathbb{P}\left(\{u(x_i)\}_{i=1}^m\right)$$
$$- \log \mathbb{P}\left(\{\widehat{u}(x_i)\}_{i=1}^n\right) \quad (24)$$

Given $\epsilon_i \sim \mathcal{N}(0, \sigma^2)$ and $\{\epsilon_i\}_{i=1}^n$ is a multivariate Gaussian, then $\{\widehat{u}(x_i)\}_{i=1}^n \Big| \{u(x_i)\}_{i=1}^n$ is a shifted multivariate Gaussian with mean $\{u(x_i)\}_{i=1}^n$ translated from the original multivariate Gaussian $\{\epsilon_i\}_{i=1}^n$. Due to the translation invariance property of Gaussian distribution, We have :

$$\log \mathbb{P}\left(\{\widehat{u}(x_i)\}_{i=1}^n \Big| \{u(x_i)\}_{i=1}^n\right) = \log \mathbb{P}\left(\{\epsilon_i\}_{i=1}^n\right) = -\frac{\sum_{i=1}^n \|\widehat{u}(x_i) - u(x_i)\|^2}{2\sigma^2} - \frac{n}{2}\log(2\pi\sigma^2) \quad (25)$$

We notice $m > n$ and $\{\widehat{u}(x_i)\}_{i=i}^n$ only depends on $\{u(x_i)\}_{i=1}^n$, and doesn't depend on $\{u(x_i)\}_{i=n+1}^m$. Thus $\log \mathbb{P}\left(\{\widehat{u}(x_i)\}_{i=1}^n \Big| \{u(x_i)\}_{i=1}^m\right) = \log \mathbb{P}\left(\{\widehat{u}(x_i)\}_{i=1}^n \Big| \{u(x_i)\}_{i=1}^n\right)$.

For evaluating $\log \mathbb{P}\left(\{u(x_i)\}_{i=1}^m\right)$, which is the second part on the right-hand side of Eq. 24, we can efficiently calculate it with the likelihood estimation tool described above. The third part on the right hand side of Eq. 24 ($\log \mathbb{P}\left(\{\widehat{u}(x_i)\}_{i=1}^n\right)$) represents the evidence and is constant. Thus the posterior distribution of Eq 24 can be simplified as:

$$\log \mathbb{P}\left(\{u(x_i)\}_{i=1}^m \Big| \{\widehat{u}(x_i)\}_{i=1}^n\right) = -\frac{\sum_{i=1}^n \|\widehat{u}(x_i) - u(x_i)\|^2}{2\sigma^2} + \log \mathbb{P}\left(\{u(x_i)\}_{i=1}^m\right) + C \quad (26)$$

Where the constant $C = -\frac{n}{2}\log(2\pi\sigma^2) - \log \mathbb{P}\left(\{\widehat{u}(x_i)\}_{i=1}^n\right)$. Given the closed-form posterior distribution, we adopt SGLD (Welling & Teh, 2011) to efficiently sample from the posterior, and then derive statistical features of interest, e.g. mean and variance, from the posterior samples. More specifically, we implement the posterior sampling strategy developed by Shi et al. (2024a), which suggests that given an invertible framework, sampling within the Gaussian process space (where the Gaussian measure $\nu_0$ is defined) and then mapping to the data function space (where data measure $\nu_1$ defined) yields better performance compared to direct sampling in the data function space. In all experiments, we use the `dopri5` ODE solver provided by `torchdiffeq` Chen et al. (2019) with `atol=1e-5` and `rtol=1e-5`. Detailed posterior sampling algorithm is provided in Appendix A.5

## 5 EXPERIMENTS

In this section, we demonstrate the superior regression performance compared to several baselines across a variety of function datasets, including both Gaussian and highly non-Gaussian Process. As baselines, we employ standard Gaussian Process Regression (Williams & Rasmussen, 2006), Deep GPs (Salimbeni & Deisenroth, 2017; Jankowiak et al., 2020), Neural Processes (Kim et al., 2019; Garnelo et al., 2018), and Operator Flow (Shi et al., 2024a).

For our function dataset, we analyze: (1) Gaussian and non-Gaussian with known posterior, including 1D Gaussian Processes, 2D Gaussian Random Fields (GRF), and 1D Truncated Gaussian Processes (TGP). (2) Highly non-Gaussian process datasets with unknown posterior, such as those derived from Navier-Stokes equations, black hole dataset from expensive Monte Carlo simulation, and 2D Signed Distance Functions extracted from MNIST digits (MNIST-SDF) (Sitzmann et al., 2020). During regression, we assume that the prior $\mathcal{G}_\theta$ is always successfully trained and remains frozen. Details about the learning process for priors are provided in the Appendix A.6.

**Gaussian Processes.** This experiment replicates the results of classical GPR, wherein the posterior distributions are precisely known in a closed form. The process involves generating a single new realization from the data measure $\nu_1$. We then select observations at 6 randomly chosen positions, incorporating a predefined noise level. The posterior is inferred across 128 positions, which includes estimating noise-free values at the observation points. We evaluate our results with two commonly used quantities in the GP literature (1) Standardized Mean Squared Error (SMSE) that normalizes the mean squared error by the variance of the ground truth; and (2) Mean Standardized Log Loss (MSLL), originally introduced by Williams & Rasmussen (2006), defined as:

$$-\log p(y_* | \{\widehat{u}(x_i)\}_{i=1}^n, x_*) = \frac{1}{2}\log(2\pi\sigma_*^2) + \frac{(y_* - \bar{y})^2}{2\sigma_*^2} \quad (27)$$

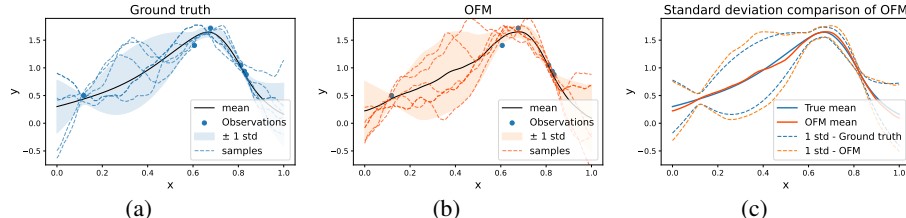

Figure 2: OFM regression on GP data. (a) Ground truth GP regression with observed data and predicted samples. (b) OFM regression with observed data and predicted samples. (c) Standard deviation comparison between true GP and OFM predictions.

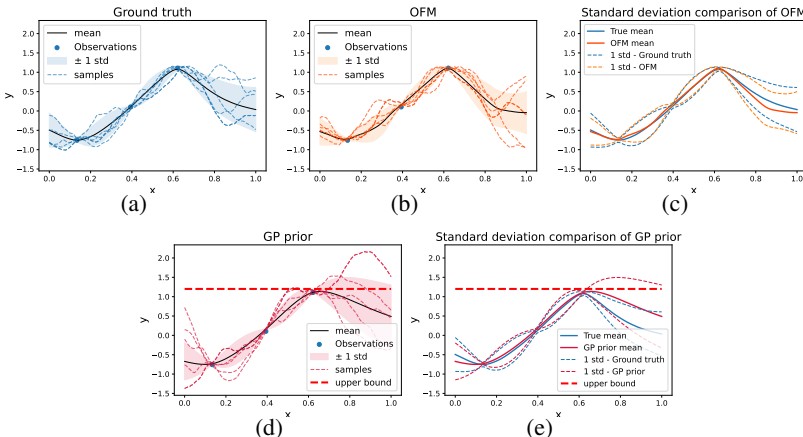

Figure 3: OFM regression on TGP data. (a) Ground truth TGP regression with observed data and predicted samples. (b) OFM regression with observed data and predicted samples. (c) Standard deviation comparison between true TGP and OFM predictions. (d) prior GP regression with observed data and predicted samples. (e) Standard deviation comparison between true TGP and GP prior predictions.

where $\{\widehat{u}(x_i)\}_{i=1}^{n}$ represents observations, $x_*, y_*$ indicate the new positions queried, and the test data (true posterior samples). Meanwhile, $\bar{y}, \sigma_*^2$ are predicted mean and variances from the model. We average out SMSE and MSLL over a test dataset contains 1000 true GP posterior samples for all models. The performance of each model is detailed in Table 1. From Fig. 2, the regression with OFM matches the analytical solution very well and provides realistic posterior samples.

**Truncated Gaussian Processes.** In this experiment, we analyze the regression performance of OFM for tractable non-Gaussian processes. Specifically, we work on truncated Gaussian Process (Swiler et al., 2020; Shi et al., 2024a), which constrains the function amplitude within a specified range. This is achieved by applying a sampling-rejection strategy on samples from the GP prior. We set the bounds of our TGP to $[-1.2, 1.2]$ and perform regression using observations only at three points, while estimating the posterior across 128 points. Subsequently, we sample 1000 true TGP posteriors from the GP prior to calculate the mean and standard deviation. Traditional metrics like MSLL and SMSE, which assume a Gaussian posterior, are not suitable for TGP. Therefore, we evaluate performance using the mean squared error for both the predicted mean and standard deviation. The results are reported in Table. 1, and illustrated in Fig. 3. OFM accurately learns the specified bounds and provides accurate estimations of mean and standard deviation, along with realistic posterior samples. In contrast, directly applying GP regression exceeds the bounds and yields unrealistic posterior samples.

**Gaussian Random Fields.** Similar to the 1D GP example, we extend our regression analysis to 2D GRF. As shown in Fig. 5 and detailed in Table 1, OFM provide accurate posterior estimation. The relative error shown in Fig. 5 is the absolute error normalized by the maximum absolute value of the mean prediction derived from the ground truth GP regression.

**Navier-Stokes, Black hole and MNIST-SDF datasets.** We collected a 2D Navier-Stokes dataset and applied OFM for the regression. Unlike Gaussian Process, where MSLL and SMSE score serve as standard benchmarks, evaluating the performance of models on general non-Gaussian processes presents a significant challenge due to the difficulty or impossibility of determining the true posterior and lack of benchmarks. Therefore, we present the predicted mean, and a posterior sample in Fig 1 for visual comparison with the ground truth. The predicted mean, along with the posterior sample, are closely aligned with the ground truth. In contrast, traditional GP regression failed to accurately capture the dynamics of the Navier-Stokes data. In Fig. 4, we conduct a similar analysis using a simulated black hole dataset. Here, OFM provides a more realistic mean and posterior sample that capture the density and swirling patterns of the black hole. Once again, GPR fails to capture these key statistics. Next, we observe similar outcomes when applying OFM to the MNIST-SDF example (Fig 7), where OFM correctly recognizes the number "7" while GPR does not.

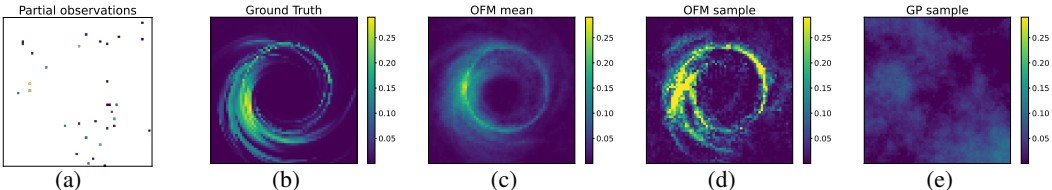

Figure 4: OFM regression on black hole data with resolution $64 \times 64$. (a) 32 random observations. (b) Ground truth sample. (c) Predicted mean from OFM. (d) One posterior sample from OFM. (e) One posterior sample from best fitted GP.

| Dataset $\rightarrow$ | 1D GP | | 2D GRF | | 1D TGP | |
|---|---|---|---|---|---|---|
| Algorithm $\downarrow$ Metric $\rightarrow$ | SMSE | SMLL | SMSE | SMLL | $\mu$ | $\sigma$ |
| GP prior | - | - | - | - | $6.4 \cdot 10^{-2}$ | $1.6 \cdot 10^{-2}$ |
| OpFlow | $5.0 \cdot 10^{-1}$ | $2.0 \cdot 10^{-1}$ | $1.4 \cdot 10^{-1}$ | $\mathbf{1.1 \cdot 10^{-1}}$ | $1.3 \cdot 10^{-2}$ | $3.9 \cdot 10^{-3}$ |
| NP | $6.1 \cdot 10^{-1}$ | $4.5 \cdot 10^{0}$ | $1.7 \cdot 10^{-1}$ | $2.1 \cdot 10^{0}$ | $1.0 \cdot 10^{-1}$ | $1.9 \cdot 10^{-2}$ |
| ANP | $5.1 \cdot 10^{-1}$ | $9.8 \cdot 10^{-1}$ | $1.6 \cdot 10^{-1}$ | $1.1 \cdot 10^{0}$ | $1.4 \cdot 10^{-1}$ | $1.7 \cdot 10^{-2}$ |
| DGP | $4.1 \cdot 10^{-1}$ | $6.8 \cdot 10^{-2}$ | $1.8 \cdot 10^{0}$ | $4.2 \cdot 10^{0}$ | $4.9 \cdot 10^{-1}$ | $1.4 \cdot 10^{-2}$ |
| DSPP | $4.7 \cdot 10^{-1}$ | $6.5 \cdot 10^{0}$ | $1.9 \cdot 10^{-1}$ | $6.6 \cdot 10^{0}$ | $1.1 \cdot 10^{-2}$ | $1.3 \cdot 10^{-2}$ |
| OFM | $\mathbf{4.1 \cdot 10^{-1}}$ | $\mathbf{5.5 \cdot 10^{-2}}$ | $\mathbf{1.3 \cdot 10^{-1}}$ | $1.6 \cdot 10^{-1}$ | $\mathbf{5.2 \cdot 10^{-3}}$ | $\mathbf{9.5 \cdot 10^{-4}}$ |

Table 1: Comparison of OFM with baseline models: GPR; OpFlow (Shi et al., 2024a); Neural Processes ( Garnelo et al. (2018), NP); Attentive NP ( Kim et al. (2019), ANP); Deep variational GP ( Salimbeni & Deisenroth (2017), DGP); Deep Sigma Point Process ( Jankowiak et al. (2020), DSSP); Datasets contain 1D GP, 2D GRF, and 1D TGP examples. Metrics SMSE and SMLL used for 1D GP and 2D GRF example. Mean squared error for the predicted mean ($\mu$) and predicted standard deviation ($\sigma$) are used for TGP example. Performance of GP regression for 1D GP and 2D GRF are removed (marked with '$-$'), which are taken as the ground truth. Best performance in bold.

## 6 CONCLUSION

In this paper, we proposed Operator Flow Matching (OFM) for stochastic process learning, which generalizes finite-dimensional marginal optimal transport flow matching model to infinite-dimensional function space. OFM efficiently computes the probability density for any finite collection of points and supports mathematically tractable functional regression. We extensively tested OFM across a diverse range of datasets, including those with closed-form GP and non-GP data, as well as highly non-GP such as Navier-Stokes and black hole data. In comparative evaluations, OFM consistently outperformed all baseline models, establishing new standards in stochastic process learning and regression.

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

# A APPENDIX

## A.1 DERIVATION OF EQ. 9 IN SECTION 3.3

In this part, we show the detailed derivation of Eq. 9. In Flow Matching, the variable $z$ is chosen as a single data point $u_1 \sim q_1$, and $u_0 \sim \mathcal{N}(\mathbf{0}, K(\{x_1, x_2, \ldots, x_n\}))$. Considering the class of Gaussian conditional probability paths

$$p_t(u_t|u_1) = \mathcal{N}(u_t|\mu_t(u_1), \sigma_t(u_1)^2 K(\{x_1, x_2, \ldots, x_n\})) \tag{28}$$

With conditional flow $\phi_t(u_t|u_1) = \sigma_t u_0 + \mu_t$. Specially, we choose $\mu_t = tu_1$ and $\sigma_t = 1 - (1-\sigma)t$, where $\sigma > 0$ is a small constant. From Eq. 10 (or Theorem 3 of Lipman et al. (2023)), a vector that defines the conditional flow is :

$$\mathcal{G}_t(u_t|u_1) = \frac{\sigma_t'}{\sigma_t}(u_t - \mu_t) + \mu_t'(u_1) \tag{29}$$

Then we can derive a closed-form expression for both the conditional probability and corresponding vector field (Tong et al., 2024) by plug in $\mu_t$ and $\sigma_t$ into Eq. 28 and Eq. 29

$$p_t(u_t|u_1) = \mathcal{N}(u_t|tu_1, (t\sigma - t + 1)^2 K(\{x_1, x_2, \ldots, x_n\})) \tag{30}$$

$$\mathcal{G}_t(u_t|u_1) = \frac{-(1-\sigma)}{1-(1-\sigma)t}(u_t - tu_1) + (u_1) = \frac{u_1 - (1-\sigma)u_t}{1-(1-\sigma)t} \tag{31}$$

Now, let's check the boundary conditions. At $t = 0$,

$$p_0(u_t|u_1) = \mathcal{N}(u_t|0, K(\{x_1, x_2, \ldots, x_n\})) = q_0 \tag{32}$$

At $t = 1$,

$$p_1(u_t|u_1) = \mathcal{N}(u_t|u_1, \sigma^2 K(\{x_1, x_2, \ldots, x_n\})) \xrightarrow{\sigma \to 0} \delta_{u_1}(u_t) \tag{33}$$

Eq. 32 and Eq. 33 describe how we interpolate between $q_0$ and $\delta_{u_1}(u_t)$, consistent with those defined in (Lipman et al., 2023). From Eq. 5, we have $p_1(u_1) = \int p_1(u_t|u_1)q_1(u_1)du_1 = q_1$ and $p_0(u_0) = \int q_0(u_0)q_1(u_1)du_1 = q_0$, which show boundary conditions are satisfied.

## A.2 MODEL STOCHASTIC PROCESS WITH INFINITE-DIMENSIONAL FLOW MATCHING VIA KOLMOGOROV EXTENSION THEOREM

Consider a Gaussian Process $\mathcal{P} : \mathcal{X} \to \mathcal{Y}$, for a finite sequence or set $\{x_1, x_2, \ldots x_n\}$ with $x_i \in \mathcal{X}$, we have $\{a(x_1), a(x_2), \ldots a(x_n)\} = \mathcal{P}(\{x_1, x_2, \ldots x_n\})$ as a multivariate Gaussian distribution. Follow the definition in Section 3.1, we define an operator $\mathcal{G}$ and for any finite set, we have

$$\{u(x_1), u(x_2), \ldots, u(x_n)\} = \mathcal{G}(\{a(x_1), a(x_2), \ldots, a(x_n)\})$$

With the with abuse of notation $\mathbb{P}(u(x))$ denotes the density of $u(x)$ at point $x$, same for $\mathbb{P}(a(x))$, then

$$\mathbb{P}(\{u(x_1), u(x_2), \ldots, u(x_n)\}) = \mathbf{J}\mathcal{G}\Big|_{\{a(x_1), a(x_2), \ldots, a(x_n)\}} \mathbb{P}(\{a(x_1), a(x_2), \ldots, a(x_n)\}). \tag{34}$$

where $\mathbf{J}\mathcal{G}\Big|_{\{a(x_1), a(x_2), \ldots, a(x_n)\}}$ is the Jacobian of the map from the collection of random variables $\{a(x_1), a(x_2), \ldots, a(x_n)\}$ at points $\{x_1, x_2, \ldots, x_n\}$ to random variables $\{u(x_1), u(x_2), \ldots, u(x_n)\}$. According to the Kolmogorov Extension Theorem (Kolmogorov & Bharucha-Reid, 2018), to establish that a valid stochastic process $\mathcal{Q}$, which has $\mathbb{P}(\{u(x_1), u(x_2), \ldots, u(x_n)\})$ as its finite dimensional distributions, it is essential to demonstrate that such a joint distribution satisfies the following two consistency properties:

**Permutation invariance.** For any permutation $\pi$ of $\{1, \cdots, n\}$, the joint distribution should remain invariant when elements of $\{x_1, \cdots, x_n\}$ are permuted, such that

$$\mathbb{P}(\{u(x_1), u(x_2), \ldots, u(x_n)\}) = \mathbb{P}(\{u(x_{\pi(1)}), u(x_{\pi(2)}), \ldots, u(x_{\pi(n)})\}) \tag{35}$$

**Marginal Consistency.** This principle specifies that that if a portion of the set is marginalized, the marginal distribution will still align with the distribution defined on the original set, such that for $m \geq n$

$$\mathbb{P}\left(\{u(x_1), u(x_2), \ldots, u(x_n)\}\right) = \int \mathbb{P}\left(\{u(x_1), u(x_2), \ldots, u(x_m)\}\right) du(x_{n+1}) \cdots du(x_m) \quad (36)$$

The permutation invariance property is naturally upheld when utilizing operator, as there is no inherent order among the elements in the set $\{x_1, x_2, \ldots, x_n\}$. Furthermore, the marginal consistency property is also maintained due to the definition of operator $\mathcal{G}$ (see Eq. 34), which ensures that $\mathbb{P}\left(\{u(x_1), u(x_2), \ldots, u(x_n)\}\right)$ is closed under marginalization. While verifying that $\mathcal{Q}$ constitutes a valid induced stochastic process is straightforward given the $\mathcal{G}$, approximating the $\mathcal{G}$ with a neural operator with induced Jacobian $\mathbf{J}\mathcal{G}\big|_{\{a(x_1), a(x_2), \ldots, a(x_n)\}}$ for any set $\{x_1, x_2, \cdots x_n\}$ is non-trivial and depends highly on the model used. In this study, we develop optimal-transport infinite-dimensional flow matching, which acts as a diffeomorphism that applies a transformation to a Gaussian measure characterized by a Gaussian process. The Jacobian matrix for any collection of points is determined by the integrating the divergence of learnt vector field as stated in Eq. 21

### A.3 EXAMPLE OF POSTERIOR SAMPLES

In this section, we initially present the regression result of OFM in another additional N-S scenario, as illustrated in Fig 6. Subsequently, we display more posterior samples used in the 2D regression examples. As depicted in Fig 8, 9, 10, OFM successfully generates realistic posterior samples that are consistent with the ground truth and demonstrate appropriate variability. In contrast, GP Regression fails to produce explainable posterior samples.

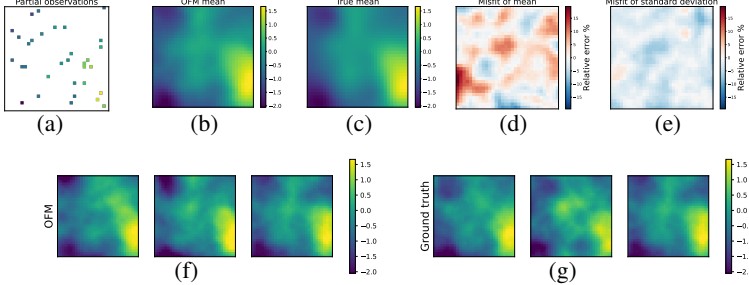

Figure 5: OFM regression on GRF data with resolution $32 \times 32$. (a) 32 random observations. (b) Predicted mean from OFM. (c) Ground truth mean from GP regression. (d) Misfit of the predicted mean. (e) Misfit of predicted standard deviation. (f) Predicted samples from OFM. (g) Predicted samples from GPR.

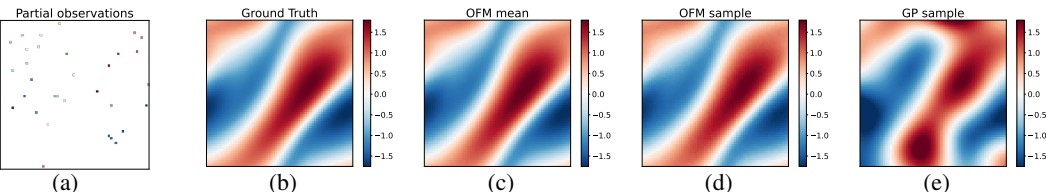

Figure 6: OFM regression on Navier-Stokes functional data with resolution $64 \times 64$. (a) 32 random observations. (b) Ground truth sample (c) Predicted mean from OFM. (d) One posterior sample from OFM. (e) One posterior sample from best fitted GP.

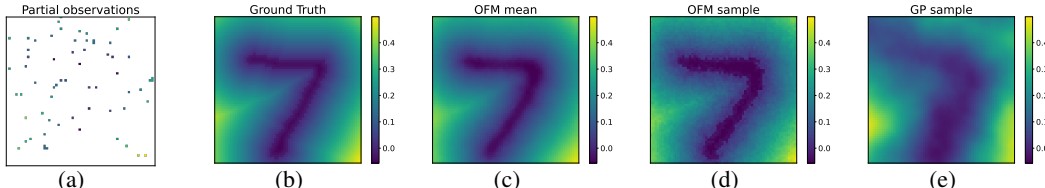

Figure 7: OFM regression on MNIST-SDF with resolution $64 \times 64$. (a) 64 random observations. (b) Ground truth sample. (c) Predicted mean from OFM. (d) One posterior sample from OFM. (e) One posterior sample from best fitted GP.

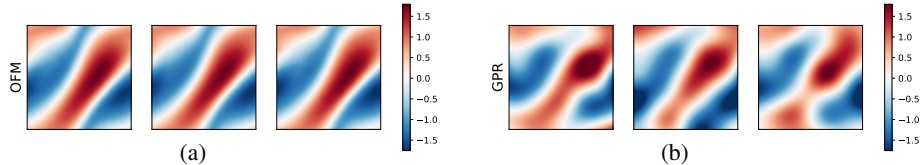

Figure 8: OFM regression on NS data. (a) Posterior samples from OFM. (b) Posterior samples from GPR.

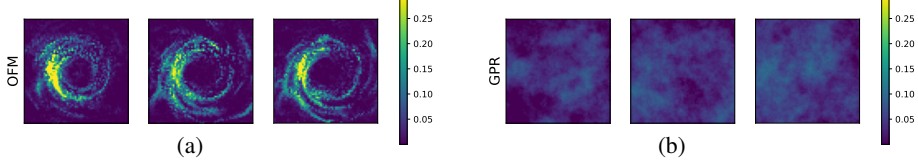

Figure 9: OFM regression on black hole data. (a) Posterior samples from OFM. (b) Posterior samples from GPR.

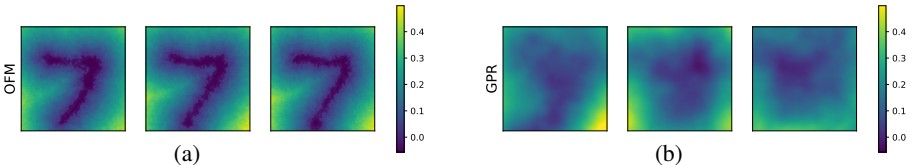

Figure 10: OFM regression on MNIST-SDF data. (a) Posterior samples from OFM. (b) Posterior samples from GPR.

### A.4    CO-DOMAIN FUNCTIONAL REGRESSION WITH OFM

In this section, we expand our regression framework to accommodate co-domain settings, as many function datasets feature a co-domain dimension greater than one. For example, earthquake waveform data commonly include three directional components, leading to a three-dimensional co-domain. Similarly, the velocity field in fluid dynamics usually features three directional components, also resulting in a dimension of co-domain of three.

We illustrate this extension through a 2D GRF example with a co-domain of 3 (channel dimension of 3). In learning the prior, we define the reference measure ($\mu_0$) as a joint measure (Wiener measure) of three identical but independent Gaussian measures while the target measure ($\mu_1$) is another Wiener measure. We keep all other parameters unchanged as those described in the 2D GRF regression tasks, with the only modification being an increase in the channel dimension from one to three. After training the prior (training detail provided in Appendix A.6), and provided 32 random observations across the three channels at co-locations, we then perform regression with OFM across these channels jointly. As demonstrated in Fig 11, OFM accurately estimate the mean and uncertainty across three channels.

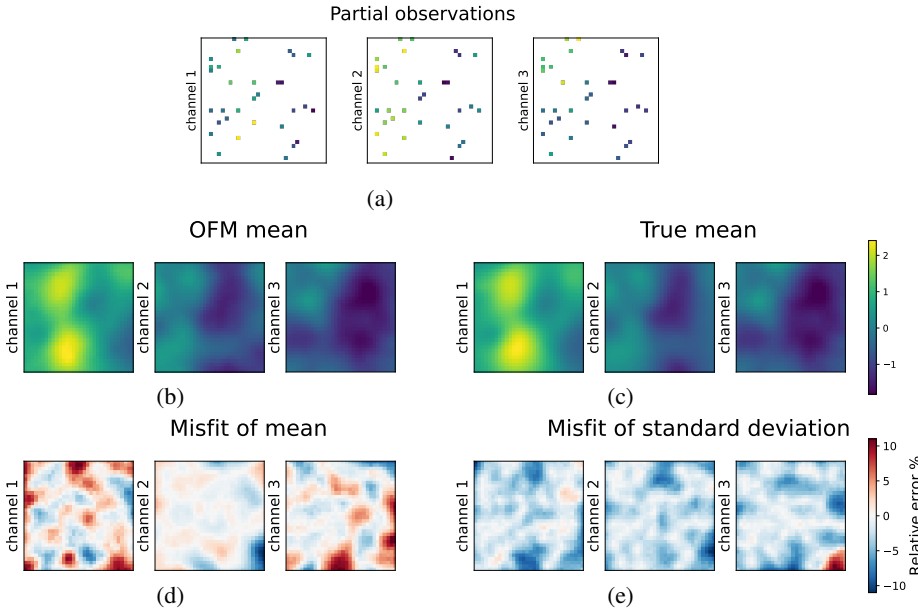

Figure 11: OFM regression on co-domain GRF data with resolution 32x32. (a) 32 random observations at co-locations. (b) Predicted mean from OFM. (c) Ground truth mean from GP regression. (d) Misfit of the predicted mean. (e) Misfit of predicted standard deviation.

### A.5    POSTERIOR SAMPLING WITH STOCHASTIC GRADIENT LANGEVIN DYNAMICS

In this section, we describe how to sample from posterior distribution with SGLD. We denote logarithmic posterior distribution (Eq. 26) as $\log \mathbb{P}_\theta$ and denote a set of posterior samples as $\{u_\theta^t\}_{t=1}^N$, where each $u_\theta^t$ is defined on a collection of point $\{x_i\}_{i=1}^m$.

By following the standard SGLD pipeline as described by Welling & Teh (2011), we can obtain a set of $N$ posterior samples $\{u_\theta^t\}_{t=1}^N$. However, SGLD is known to be sensitive to the choice of regression parameters and can become trapped in local minima, leading to convergence issues, especially in regions of high curvature (Li et al., 2015). To mitigate these challenges, Shi et al. (2024a) proposed that within an invertible framework, drawing a posterior sample $u_\theta^t$ is equivalent to drawing a sample $a_\theta^t$ in Gaussian space, since $u_\theta^t$ uniquely defines $a_\theta^t$ and vice versa. This approach can stabilize

the posterior sampling process and is less sensitive to the regression parameters due to the inherent smoothness of the Gaussian process. Additionally, Shi et al. (2024a) suggests starting from maximum a *posteriori* (MAP) estimate of $a_\theta^t$, denoted as $\overline{a_\theta}$, which can reduces the number of burn-in terations needed in SGLD. We adopt the same sampling strategy and refer readers to the detailed discussion in Shi et al. (2024a). The algorithm is reported in Algorithm 1

When the size of observations or context points ($\{\widehat{u}(x_i)\}_{i=1}^n$) is 0, sampling from the posterior degrades to sampling from the prior, the results of which are presented in the subsequent section.

---

**Algorithm 1** Posterior sampling with SGLD

---

1: **Input and Parameters:** Logarithmic posterior distribution $\log \mathbb{P}_\theta$, temperature $T$, learning rate $\eta_t$, MAP $\overline{a}_\theta$, burn-in iteration $b$, sampling iteration $t_N$, total iteration $N$.
2: **Initialization**: $a_\theta^0 = \overline{a}_\theta$
3: **for** $t = 0, 1, 2, \ldots, N$ **do**
4:     Compute gradient of the posterior: $\nabla_{a_\theta} \log \mathbb{P}_\theta$
5:     Update $a_\theta^{t+1}$: $a_\theta^{t+1} = a_\theta^t + \frac{\eta_t}{2} \nabla \log \mathbb{P}_\theta + \sqrt{\eta_t T} \mathcal{N}(0, I)$
6:     **if** $t \geq b$ **then**
7:         Every $t_N$ iterations: obtain new sample $a_\theta^{t+1}$, and corresponding $u_\theta^{t+1}$
8:     **end if**
9: **end for**

---

### A.6 Prior Learning With OFM

In this part, we elaborate the prior learning process and the evaluation of performance. We employ Matern kernel to construct the reference GP and to prepare training datasets for 1D GP, 2D GRF, and 1D TGP. We have set the kernel length $l = 0.01$ with a smoothness factor $\zeta = 0.5$ for all reference GPs. OFM maps the GP samples from reference GPs to data samples and is resolution-invariant, which means OFM can be trained with functions at any resolution and evaluated at any resolution.

**1D GP dataset.** We choose $l = 0.3$ and $\zeta = 1.5$ and generate $20,000$ training samples on domain $[0, 1]$ with a fixed resolution of 256. We use autocovariance and histogram of point-wise value as metrics for evaluation. We evaluate OFM at several different resolutions shown Fig 12, 13, 14, which demonstrate OFM's excellent capability to learn the function prior.

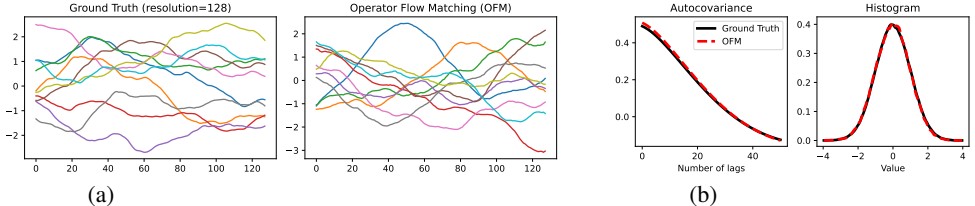

Figure 12: OFM for 1D GP prior learning, evaluated at resolution=128. (a) Random samples from ground truth and generated by OFM. (b) Autocovariance and histogram comparison

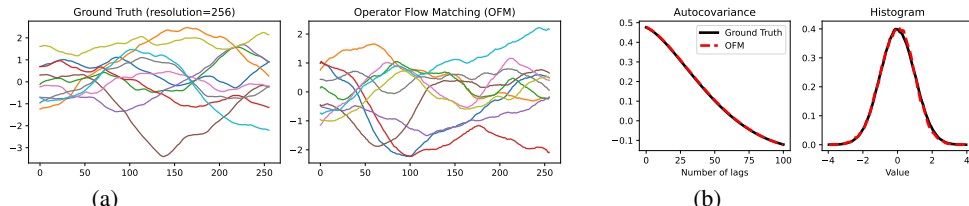

Figure 13: OFM for 1D GP prior learning, evaluated at resolution=256. (a) Random samples from ground truth and generated by OFM. (b) Autocovariance and histogram comparison

**1D TGP dataset.** We choose $l = 0.3$ and $\zeta = 1.5$ and generating $20,000$ training samples on domain $[0, 1]$ with a fixed resolution of 256. We set $[-1.2, 1.2]$ for the bounds. Results provided in Fig 15, 16, 17.

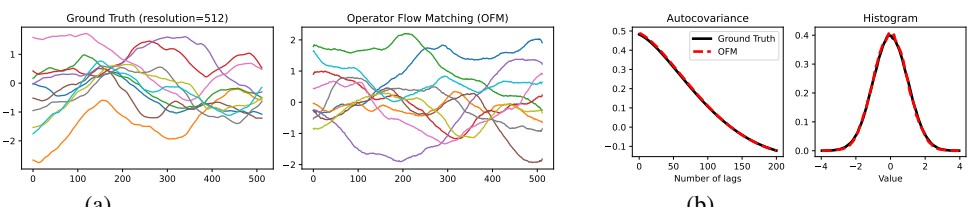

Figure 14: OFM for 1D GP prior learning, evaluated at resolution=512. (a) Random samples from ground truth and generated by OFM. (b) Autocovariance and histogram comparison

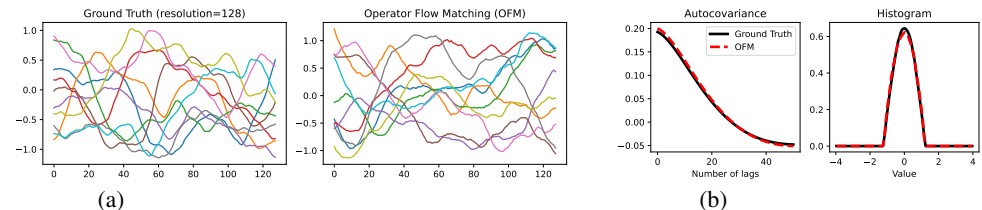

Figure 15: OFM for 1D TGP prior learning, evaluated at resolution=128. (a) Random samples from ground truth and generated by OFM. (b) Autocovariance and histogram comparison

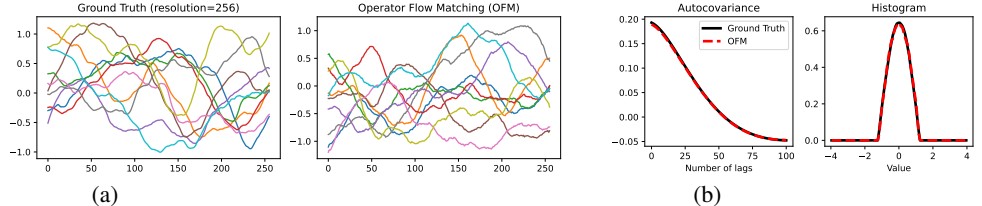

Figure 16: OFM for 1D TGP prior learning, evaluated at resolution=256. (a) Random samples from ground truth and generated by OFM. (b) Autocovariance and histogram comparison

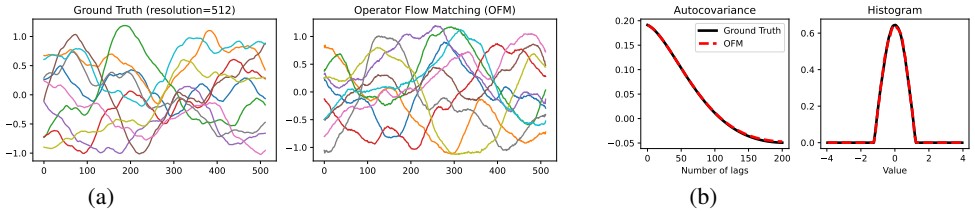

Figure 17: OFM for 1D TGP prior learning, evaluated at resolution=512. (a) Random samples from ground truth and generated by OFM. (b) Autocovariance and histogram comparison

**2D Naiver-Stokes, Black hole, MNIST-SDF datasets.** All the following 2D datasets are defined on domain $[0, 1] \times [0, 1]$ and have a resolution of $64 \times 64$. We collected a 2D Navier-Stokes dataset consisting of 20000 samples, with viscosity $= 1e - 4$. The results, including zero-shot super-resolution, are provided in Fig 18, 19. The learning of Black hole dataset, generated using expensive Monte Carlo method, is detailed in Fig 20, 21. Additionally, we trained OFM on $20,000$ MNIST-SDF samples, the outcomes are illustrated in Fig 22, 23.

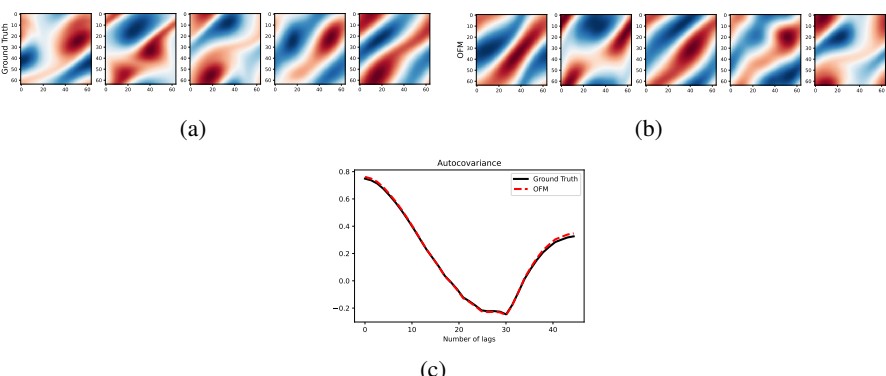

(a)                  (b)

(c)

Figure 18: OFM for 2D N-S prior learning, evaluated at resolution=$64 \times 64$. (a) Random samples from ground truth. (b) Random samples generated by OFM. (c) Autocovariance comparison

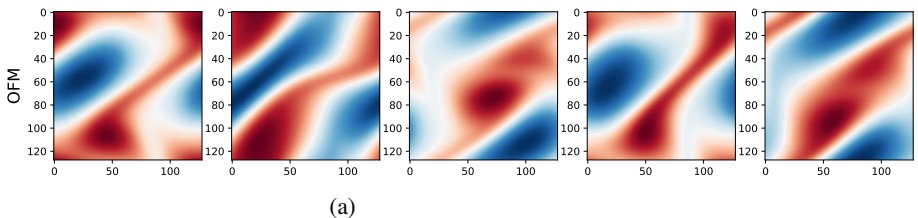

(a)

Figure 19: OFM for 2D N-S prior learning, evaluated at $128 \times 128$ resolution (zero-shot super-resolution)

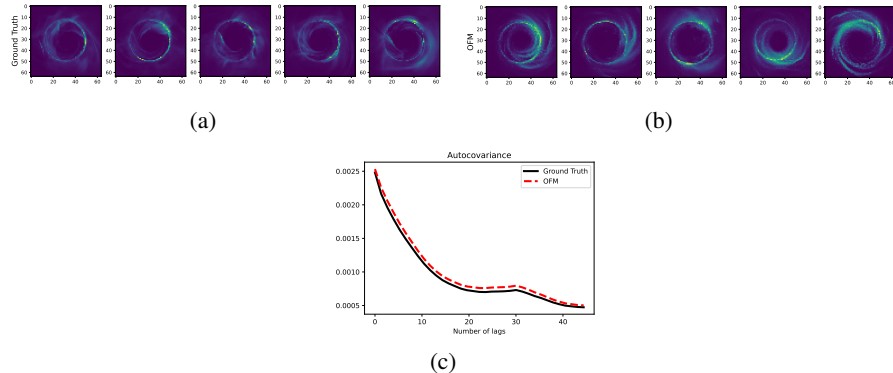

(a)                  (b)

(c)

Figure 20: OFM for 2D black hole prior learning, evaluated at resolution=64. (a) Random samples from ground truth. (b) Random samples generated by OFM. (c) Autocovariance comparison

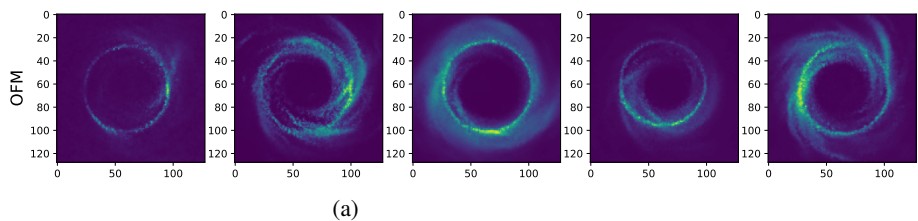

(a)

Figure 21: OFM for 2D black hole prior learning, evaluated at $128 \times 128$ resolution (zero-shot super-resolution)

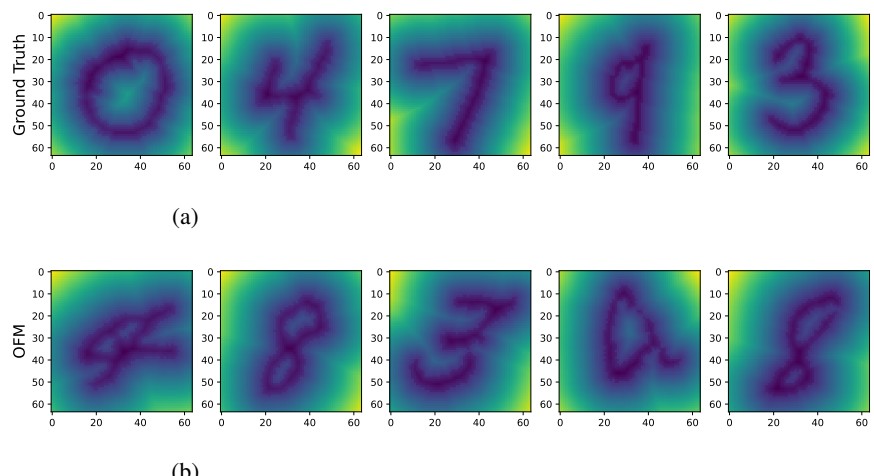

(a)

(b)

Figure 22: OFM for 2D MNIST-SDF prior learning, evaluated at $64 \times 64$ resolution. (a) Random samples from ground truth. (b) Random samples generated by OFM.

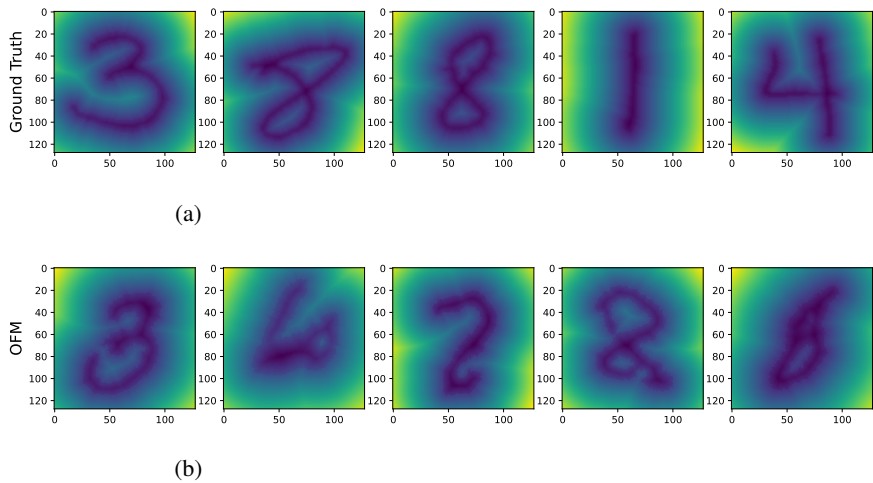

(a)

(b)

Figure 23: OFM for 2D MNIST-SDF prior learning, evaluated at $128 \times 128$ resolution. (a) Random samples from ground truth. (b) Random samples generated by OFM.

## A.7 DETAILS OF EXPERIMENTAL SETUP

In this section, we outline the details of experiments setup used in this paper. Since regression with OFM requires learning the prior first, we list the parameters used for learning the prior and regression separately. We employ FNO as the backbone, implemented using `neuraloperator` library (Li et al., 2021). All time reported in the subsequent tables are based on one computations performed using a single NVIDIA RTX A6000 (48 GB) graphics card.

Table 2 details the parameters used for training the prior. For instance, in the 1D GP prior learning experiment, the dataset consists of 20,000 samples, each with a co-domain dimension (or channel) of one. The batch size is set at 1024, and the model is trained over 500 epochs. The total training time is about 0.76 hours, and the size of the trained model is 37.1 megabytes.

Tables 3, 4, and 5 detail the parameters for SGLD sampling as described in Algorithm 1. For example, in the 1D GP regression as an example, the regression takes 40,000 iterations with a burn-in phase of 3,000 iterations. Posterior samples are collected every 10 iterations. The temperature for the injected noise during the gradient update is set at 1, and the learning rate decays exponentially from 0.005 to 0.004 (defined in Algorithm 1). We average 32 runs with the Hutchinson trace estimator to evaluate the likelihood, utilizing GPU parallel computing. The noise level, as specified in Equation 26, is 0.01 in this regression task. Then given 6 random observations, we ask for the posterior samples across 128 points. The GPU memory usage for the regression task is 4 gigabytes, with the total runtime to 4.91 hours.

| Datasets | Size of Dataset | Channels | Batch Size | Epochs | Training Time | Model Size |
|---|---|---|---|---|---|---|
| 1D GP | $2 \cdot 10^4$ | 1 | 1024 | $5 \cdot 10^2$ | 0.76 h | 37.1 MB |
| 1D TGP | $2 \cdot 10^4$ | 1 | 1024 | $5 \cdot 10^2$ | 1.24 h | 37.1 MB |
| 2D GRF | $2 \cdot 10^4$ | 1 | 256 | $5 \cdot 10^2$ | 1.14 h | 76 MB |
| 2D co-domain GRF | $2 \cdot 10^4$ | 3 | 256 | $5 \cdot 10^2$ | 1.01 h | 76 MB |
| 2D N-S | $2 \cdot 10^4$ | 1 | 256 | $5 \cdot 10^2$ | 3.79 h | 286 MB |
| 2D Black hole | $1.2 \cdot 10^4$ | 1 | 256 | $5 \cdot 10^2$ | 2.28 h | 286 MB |
| 2D MNIST-SDF | $2 \cdot 10^4$ | 1 | 256 | $5 \cdot 10^2$ | 8.31 h | 286 MB |

Table 2: Parameters used in experiments of prior learning

| Datasets | Total Iteration | Burn-in Iteration | Sampling Iterations | Temperature of Noise |
|---|---|---|---|---|
| 1D GP | $4 \cdot 10^4$ | $3 \cdot 10^3$ | 10 | 1 |
| 1D TGP | $4 \cdot 10^4$ | $3 \cdot 10^3$ | 10 | 1 |
| 2D GRF | $2 \cdot 10^4$ | $3 \cdot 10^3$ | 10 | 1 |
| 2D co-domain GRF | $2 \cdot 10^4$ | $3 \cdot 10^3$ | 10 | 1 |
| 2D N-S | $2 \cdot 10^4$ | $3 \cdot 10^3$ | 10 | 1 |
| 2D Black hole | $2 \cdot 10^4$ | $3 \cdot 10^3$ | 10 | 1 |
| 2D MNIST-SDF | $2 \cdot 10^4$ | $3 \cdot 10^3$ | 10 | 1 |

Table 3: Parameters used in regression experiments - Part A

| Datasets | Initial Learning Rate | End Learning Rate | Hutchinson Samples | Noise Level |
|---|---|---|---|---|
| 1D GP | $5 \cdot 10^{-3}$ | $4 \cdot 10^{-3}$ | 32 | $1 \cdot 10^{-2}$ |
| 1D TGP | $5 \cdot 10^{-3}$ | $4 \cdot 10^{-3}$ | 32 | $1 \cdot 10^{-3}$ |
| 2D GRF | $1 \cdot 10^{-3}$ | $8 \cdot 10^{-4}$ | 32 | $1 \cdot 10^{-2}$ |
| 2D co-domain GRF | $1 \cdot 10^{-3}$ | $8 \cdot 10^{-4}$ | 16 | $1 \cdot 10^{-2}$ |
| 2D N-S | $3 \cdot 10^{-3}$ | $2 \cdot 10^{-3}$ | 8 | $1 \cdot 10^{-3}$ |
| 2D Black hole | $5 \cdot 10^{-3}$ | $4 \cdot 10^{-3}$ | 8 | $1 \cdot 10^{-3}$ |
| 2D MNIST-SDF | $5 \cdot 10^{-3}$ | $4 \cdot 10^{-3}$ | 8 | $1 \cdot 10^{-3}$ |

Table 4: Parameters used in regression experiments - Part B

| Datasets | Number of Observations | Inquired Grids | GPU Memory | Running Time |
|---|---|---|---|---|
| 1D GP | 6 | 128 | 4 GB | 4.91 h |
| 1D TGP | 3 | 128 | 4 GB | 5.42 h |
| 2D GRF | 32 | $32 \times 32$ | 22 GB | 9.70 h |
| 2D co-domain GRF | 32 | $32 \times 32$ | 31 GB | 5.05 h |
| 2D N-S | 32 | $64 \times 64$ | 44 GB | 13.65 h |
| 2D Black hole | 32 | $64 \times 64$ | 44 GB | 13.37 h |
| 2D MNIST-SDF | 64 | $64 \times 64$ | 44 GB | 9.41 h |

Table 5: Parameters used in regression experiment - Part C

## A.8 DETAILED ANALYSIS OF OFM AND COMPARISON WITH EXISTING METHODS

In this section, we elaborate the connection and difference with pervious work, highlight contributions and potential limitations of our work. The regression with OFM involves a two-steps process: (i) learning a prior on function space, and (ii) sampling from the posterior given observations. Consequently, the OFM framework has connections with both generative models on function space and the models developed for functional regression. In the following, we provide a comprehensive comparative analysis with related models and baselines, including operator flow (OpFlow) (Shi et al., 2024a), conditional optimal transport flow matching (COT-FM) (Kerrigan et al., 2024), neural processes (NPs) (Garnelo et al., 2018; Dutordoir et al., 2023)

**Comparison with OPFLOW.** OPFLOW introduces invertible neural operators, which generalizes RealNVP (Dinh et al., 2017) to function space and maps any collection of points sampled from a GP to a new collection of points in the data space, using the maximum likelihood principle (Shi et al., 2024a). This method captures the likelihood of any collection of point consistently as the resolution increases and allows for UFR using SGLD. Despite these advantages, the requirement for an invertible neural operator brings training and expressiveness challenges. On the contrary, OFM adopts a simulation-free ODE framework for prior learning, which offers enhanced expressiveness and ensures training stability through a simple regression objective while avoiding using the invertible neural operator. In addition, OFM proposes a non-trivial extension of UFR to the simulation-free ODE framework. These improvements render OFM a more practical solution for challenging functional regression tasks.

**Comparison with COT-FM.** COT-FM (Kerrigan et al., 2024) proposes a conditional generalization of Benamou-Brenier Theorem (Benamou & Brenier, 2000), formulating a conditional optimal transport plan that applicable for both Euclidean and Hilbert space. In contrast, OFM employs an unconditional optimal transport plan in Hilbert space based on dynamic Kantorovich formulation (Chizat et al., 2018). The advantage of COT-FM lies in its ability to flexibly incorporate specific conditions tailored for conditional generative tasks. However, COT-FM is not suitable for functional regression tasks due to: (i) COT-FM is contingent upon both the reference and target being influenced by conditions, and the vector field learnt is triangular, designed to transport jointly the coupling of a reference measure and a condition measure. In UFR setting, the learnt prior is required to be unconditioned, (ii) the coupling with condition measure typically prevents inducing valid stochastic process, even when the reference measure is a Gaussian measure, (iii) cannot provide point evaluation of probability density. Last, We should notice, the development of OFM is different and independent of COT-FM, the former with a focus on stochastic process learning and Bayesian functional regression.

**Comparison with NPs.** NPs were developed to address the computational and restrictive prior challenges of Gaussian Processes, utilizing neural networks for efficiency (Garnelo et al., 2018). However, several recent studies have discussed the drawbacks in the formulation of NPs, raising concerns that NPs might not learn the underlying function distribution (Rahman et al., 2022; Dupont et al., 2022; Shi et al., 2024a).

Notably, NPs treats the point cloud data as a set of values, ignoring the metric space of the data (Dupont et al., 2022). This can lead to misinterpretations of a function sampled at different resolutions as distinct functions (Appendix A.1 of (Rahman et al., 2022)). Furthermore, NPs rely on encoding input data into finite-dimensional, Gaussian-distributed latent variables before projecting these into an infinite-dimensional space. This process tends to lose consistency at higher resolutions. Moreover, the Bayesian framework underpinning NPs focuses on point sets rather than the functions themselves, leading to a dilution of prior information with increasing data points.

In recent study, diffusion-based variants of NPs (NDP) (Dutordoir et al., 2023), was proposed to leverage the expressiveness of diffusion models (Ho et al., 2020; Song et al., 2021). Nonetheless, the formulation of NDP does not address the aforementioned issues of NPs and introduces two significant problems: (i) NDP fails to induce a valid stochastic process as it does not satisfy the marginal consistency criterion required by Kolmogorov Extension Theorem (Kolmogorov & Bharucha-Reid, 2018), and (ii) it relies on uncorrelated Gaussian noise for denoising, which is not applicable in function spaces (Lim et al., 2023). Oppositely, OFM establishes a more theoretically sound framework by rigorously defining learning within function spaces. Additionally, Bayesian functional regression within the OFM framework adheres to valid stochastic processes, offering a robust and theoretically grounded solution.

**Contribution and Limitations.** In conclusion, OFM represents the first simulation-free ODE framework designed for functional regression purpose, demonstrating superior performance over existing baselines. The theory development for generalizing flow matching to stochastic process as well as development of optimal-transport infinite-dimensional flow matching are considered as additional contributions.

Despite these advances, the current regression framework with OFM is primarily limited to low-dimensional data (1D and 2D in this study). This limitation stems from the challenges associated with learning operators for functions defined on high-dimensional domains—an area that remains underdeveloped both computationally and in terms of dataset availability (Kovachki et al., 2023). Additionally, while the time complexity for regression with OFM is $\mathcal{O}(D^2)$, the incorporation of additional components significantly increases its computational resource requirements compared to classical GP regression.

