# OpenReview forum: "Stochastic Process Learning via Operator Flow Matching"
_ICLR.cc/2025/Conference — Submitted to ICLR 2025_

### Official Review · Reviewer_RQAu · 2024-10-17

**Soundness:** 3
**Presentation:** 3
**Contribution:** 3
**Rating:** 8
**Confidence:** 4

**Summary:**

This paper propose operator flow matching (OFM) for stochastic process learning, which generalize the FM to learn functional vector field and hence we can obtain samples from infinite-dimensional function space. They also provide an efficient way for computing probability density for any finite collection of points, which enables trctable functional regression at new points. In the experiments part, they illustrate their OFM and compare it to other methods, using 1D and 2D examples.

**Strengths:**

1. The paper is well written. The idea is clear and easy to follow.
2. Extending FM to functional space can be be useful even beyond stochastic process learning.

**Weaknesses:**

Maybe the experiements part is too simple, as the dimension (in terms of both $n$ and dimension of u) is too low? There may be some computational issue for practice. Anway, this paper provides a framework for functional space extension, and these experiements are enough to illustrate the idea. Therefore, I don't take points off for this point.

**Questions:**

- Q1. the experiement setting: let's take 1D-GP experiment for example. Here $n = 6$, and use the same $(x_1,...,x_6)$ for each of 20000 trainings? If I'm correct, the current OFM can only apply to repeated trials in scientific settings (and in 2D image, apply to samples from the same location)?

- Q2. In practice, if there are many observations ($n$ is large) and dimension of $u\in \mathbb{R}^d$ is high, will FNO still feasible? Since now the dimension for OFM should be $nd$?

---

> ### Author Response · Authors · 2024-11-22
> **Official Comment by Authors Part I**
>
> We very much appreciate the positive feedback on our work and are delighted that the reviewer found our contributions useful. We have carefully considered the reviewer's concerns and made changes in the draft accordingly
>
> **Q0. Maybe the experiements part is too simple, as the dimension (in terms of both $n$ and dimension of u) is too low? There may be some computational issue for practice. Anway, this paper provides a framework for functional space extension, and these experiements are enough to illustrate the idea. Therefore, I don't take points off for this point.**
>
> This is an excellent point. We now provide functional regression results on finer grids (currently 64x64, previous 32x32) for Navier-Stokes, Black hole, and MNIST-SDF examples. We acknowledge that our OFM and regression framework are yet to advance to high dimensions as a state of deep learning progress. As the reviewer pointed out, this limitation stems from the challenges associated with learning operators for functions defined on high-dimensional domains—an area of active research, both computationally and in terms of dataset availability (kovachki et al, 2023).
>
> Additionally, while the time complexity for regression with OFM is $\mathcal{O}(D^2)$, the incorporation of additional components significantly increases its computational resource requirements compared to classical GP regression. We elaborate on this potential limitation in Appendix A.8 of the revised draft
>
>
> Reference:
>
> Nikola Kovachki, Zongyi Li, Burigede Liu, Kamyar Azizzadenesheli, Kaushik Bhattacharya, Andrew Stuart, and Anima Anandkumar. Neural Operator: Learning Maps Between Function Spaces With Applications to PDEs. Journal of Machine Learning Research, 24(89):1–97, 2023. ISSN 1533-7928. URL http://jmlr.org/papers/v24/21-1524.html.
>
> Kamyar Azizzadenesheli, Nikola Kovachki, Zongyi Li, Miguel Liu-Schiaffini, Jean Kossaifi, and
> Anima Anandkumar. Neural operators for accelerating scientific simulations and design. Nature
> Reviews Physics, 6(5):320–328, May 2024. ISSN 2522-5820. doi: 10.1038/s42254-024-00712-5. URL https://www.nature.com/articles/s42254-024-00712-5. Publisher: Nature Publishing Group.
>
> **Q1. the experiement setting: let's take 1D-GP experiment for example. Here $n=6$, and use the same $(x_1, x_2, \cdots x_6)$ for each of 20000 trainings? If I'm correct, the current OFM can only apply to repeated trials in scientific settings (and in 2D image, apply to samples from the same location)?**
>
>
> The regression framework of OFM is highly flexible; it enables GP-style regression for non-GP tasks: given observations and noise level, we inquire about the posterior distribution of at new positions. The primary distinction  between GP regression and regression with OFM is that GP prior is usually determined by the GP kernel, which is manually tuned. In contrast, in OFM, the prior of the stochastic process is learnt from the data.
>
> In the experiment mentioned by the reviewer, we first train the OFM model to learn the prior on the stochastic process and then show the regression against six arbitrary points. And yes, we need repeated trials to train the OFM model. Similar steps are also taken in GP-based regression, where historical data is used to tune the parameters of GP. However, GP is much simpler to be tuned (to a suboptimal solution).
>
>
> To elaborate more, let us first look at classical GP regression on the 1D-GP experiment. Initially, we select a GP kernel with specific parameters to define the GP prior. Then given noisy observation $\\{\widehat u(x_1), .. \widehat u(x_n)\\}$ of an unknown function drawn from the GP prior at position $\\{x_1, … x_n\\}$ , we are asked to inquire about the posterior distribution of the noise-free guesses at “m” points. That is $\mathbb{P}(\\{u(x_1), … u(x_n), .. u(x_m)\\})$ with $m>=n$. In the 1D-GP experiment, $n=6$, and $m=128$.  We should note that both $n$ and $m$ as well as positions $\\{x_1, … x_n\\}, \\{x_1, …, x_m\\}$ can vary from task to task with the same GP prior.
>
> Next, let’s examine how we perform regression with OFM for the 1D-GP experiment. Before beginning the regression, we first collect a dataset containing many 1D-GP function samples to train the prior. It is important to emphasize that this training dataset should be independent of the observations used in subsequent regression tasks (see detailed prior learning in Appendix A.6). Once trained, we will freeze the learned prior for the 1D GP experiment. All other steps remain the same as in classical GP regression
>
> Herein lies the key difference: GP regression is only accurate when we explicitly know that the potential stochastic process is GP and exactly matches the GP prior defined by the kernel. In contrast, with OFM, we don’t have these concerns as the prior of the stochastic process is learned directly from the training dataset.

---

> ### Author Response · Authors · 2024-11-22
> **Official Comment by Authors Part II**
>
> **Q2. In practice, if there are many observations ( $n$ is large) and dimension of  $u$ is high, will FNO still feasible? Since now the dimension for OFM should be $nd$?**
>
> FNO is memory and time efficient. It is known for being resolution-invariant, allowing it to be directly generalized to any resolution without re-training. The size of FNO is proportional to the number of truncated Fourier modes, and its time complexity is quasi-linear. (Li et al, 2021)
>
> However, learning high-dimensional functions (e.g., in 4D and above) still poses challenges for FNO, and deep learning in general. The difficulties stem from the complexities associated with learning operators for functions defined on high-dimensional domains, requiring massive memory for deep learning models. This is an active area of research, both computationally and in terms of dataset availability (Kovachki et al, 2023)
>
> Reference:
>
> Zongyi Li, Nikola Kovachki, Kamyar Azizzadenesheli, Burigede Liu, Kaushik Bhattacharya, Andrew Stuart, and Anima Anandkumar. Fourier Neural Operator for Parametric Partial Differential Equations, May 2021. URL http://arxiv.org/abs/2010.08895. arXiv:2010.08895
> [cs, math].
>
>
> Nikola Kovachki, Zongyi Li, Burigede Liu, Kamyar Azizzadenesheli, Kaushik Bhattacharya, Andrew Stuart, and Anima Anandkumar. Neural Operator: Learning Maps Between Function Spaces With Applications to PDEs. Journal of Machine Learning Research, 24(89):1–97, 2023. ISSN 1533-7928. URL http://jmlr.org/papers/v24/21-1524.html.

---

### Official Review · Reviewer_pMpj · 2024-10-25

**Soundness:** 3
**Presentation:** 2
**Contribution:** 1
**Rating:** 3
**Confidence:** 3

**Summary:**

The authors explore flow matching in infinite-dimensional spaces and propose a training objective to optimize a neural network-parameterized vector field (or conditional vector field). They further introduce a resolution-free Bayesian universal functional regression framework. This approach is then applied to a range of GP and non-GP regression tasks.

**Strengths:**

* The paper is well-motivated and interesting.
* The development of algorithms for infinite-dimensional flow matching and conditional flow matching is interesting.
* The authors introduce a Bayesian functional regression algorithm.

**Weaknesses:**

* In Sections 4.1 to 4.2, the authors introduce an infinite-dimensional flow matching approach, termed OFM, along with its conditional flow matching extension. Given that recent work has tackled flow matching for the conditional optimal transport problem formulated in Hilbert spaces (and hence in infinite dimensions) [1], the primary novelty here seems to center on the Bayesian UFR discussed in Section 4.3. If the proposed methods may differ from [1], further discussion compares with [1] would be appreciated.
* Despite the definition of the proposed OFM on infinite-dimensional spaces, the authors formulate the UFR problem over a finite collection of data points, assuming a density over this finite set. I understand that real-world data is always a finite number of evaluations; however, having established an infinite-dimensional framework, justifying a density function due to finite data points raises some questions. Would it not be more consistent to define the distribution as an approximation based on finite data points within this infinite-dimensional setting? In such a case, since Lesbesgue measure is not available in infinite-dimensional spaces, further discussion on defining a density function under these circumstances would be helpful. Furthermore, if this is the case, I am also curious whether the Hutchinson trace estimator proposed by the authors is well-defined in Hilbert spaces.
* In Section 5, the authors compare several methods, including GP and NP approaches, for GP regression. However, regarding the NP experiments, it appears that some baselines may be outdated. In particular, comparing with models like NDP [2], which also propose infinite-dimensional generative models, seems crucial. Demonstrating strong performance against these more recent models would be beneficial. Moreover, recent NP models tackle not only 1D problems but also more complex tasks, such as predicting full RGB-channel images from partially observed 2D domains. This image completion task is challenging, as it lacks the smooth properties typical of PDEs, yet these models show promising performance. I am curious to know if the proposed approach can also handle such complex image functions effectively.

```
[1] Kerrigan et al., Dynamical Conditional Optimal Transport through Simulation-Free Flows
[2] Dutordoir et al., Neural Diffusion Processes
```

**Questions:**

See weaknesses.

---

> ### Author Response · Authors · 2024-11-22
> **Official Comment by Authors Part I**
>
> We are delighted that the reviewer found our work well-motivated and interesting. In the following, we address the comments and explain changes made in the main text. These changes in the main text are colored in blue in the revised draft.
>
> **Q1. In Sections 4.1 to 4.2, the authors introduce an infinite-dimensional flow matching approach, termed OFM, along with its conditional flow matching extension. Given that recent work has tackled flow matching for the conditional optimal transport problem formulated in Hilbert spaces (and hence in infinite dimensions) [1], the primary novelty here seems to center on the Bayesian UFR discussed in Section 4.3. If the proposed methods may differ from [1], further discussion compares with [1] would be appreciated.**
>
> This is a great suggestion, we provide a detailed comparison in Appendix A.8, elaborating on the connection and differences between our work and COT-FM (Kerrigan et al, 2024). Notably, both OFM and COT-FM leverage optimal transport theorem, however, COT-FM is not applicable for Bayesian functional regression, we discuss this into details in Appendix A.8
>
> Reference:
>
> Gavin Kerrigan, Giosue Migliorini, and Padhraic Smyth. Dynamic Conditional Optimal Transport  through Simulation-Free Flows, May 2024. URL http://arxiv.org/abs/2404.04240. arXiv:2404.04240.
>
> **Q2. Despite the definition of the proposed OFM on infinite-dimensional spaces, the authors formulate the UFR problem over a finite collection of data points, assuming a density over this finite set. I understand that real-world data is always a finite number of evaluations; however, having established an infinite-dimensional framework, justifying a density function due to finite data points raises some questions. Would it not be more consistent to define the distribution as an approximation based on finite data points within this infinite-dimensional setting? In such a case, since Lesbesgue measure is not available in infinite-dimensional spaces, further discussion on defining a density function under these circumstances would be helpful. Furthermore, if this is the case, I am also curious whether the Hutchinson trace estimator proposed by the authors is well-defined in Hilbert spaces.**
>
> This is a great point. Initially, the connection between the finite-dimensional marginal (equipped with Lebesgue measure) and the probability measure of a stochastic process in infinite-dimensional space is defined by Kolmogorov Extension theorem (KET), which assures that if all finite-dimensional distributions (i.e., distributions of function at finite collection of points) are consistent, then a stochastic process exists that matches finite-dimensional distributions,.
>
> To understand this intuitively, consider GP regression as an example. Assuming a GP prior defined in infinite-dimensional space with its finite-dimensional marginal being multivariate Gaussian (wrt Lesbesgue measure), we are given noisy observations $\lbrace \widehat u(x_1),\widehat u(x_2),\ldots,\widehat u(x_n)\rbrace$ at the position $\\{x_1, x_2, \ldots, x_n\\}$ of a unknown function drawn from the prior (wrt Gaussian measure). We are asked to get the noise-free posterior distribution of  $\lbrace u(x_1), u(x_2),\ldots, u(x_m)\rbrace$ with $m>n$. It's important to note that there is no inherent order in the set $\\{x_1, x_2, \ldots, x_n\\}$ in GP regression, and positions of observations, as well as the positions inquired, can vary from case to case with the same GP prior.
>
> We compute the posterior over finite collection points equipped with Lebesgue measure, by KET, the GP posterior is still over functions (wrt Gaussian measure). What’s more, in classical GP regression, the observation is usually corrupted with white noise, which doesn’t exist in L2 space, but this consideration is important for practical applications.
>
> Returning to the setting with OFM, after learning the prior and given some observations, we are tasked with providing the posterior distribution at specific locations. This shifts the problem from an infinite-dimensional to a finite-dimensional setting. The Hutchinson trace estimator serves as a practical tool for providing the unbiased estimation of an induced Jacobian matrix by any finite-dimensional distribution. The learnt prior in function space remains frozen. Lastly, we have revised our narrative in Section 3.1 to enhance readability and provide a background and proof that generalizes flow matching to stochastic process via KET in appendix A.2
>
>
>
>
>
> Reference:
>
> Andre˘ı Nikolaevich Kolmogorov and Albert T Bharucha-Reid. Foundations of the theory of probability: Second English Edition. Courier Dover Publications, 2018. ISBN 0-486-82159-5

---

> > ### Author Response · Authors · 2024-11-22
> > **Official Comment by Authors Part II**
> >
> > **Q3. In Section 5, the authors compare several methods, including GP and NP approaches, for GP regression. However, regarding the NP experiments, it appears that some baselines may be outdated. In particular, comparing with models like NDP [2], which also propose infinite-dimensional generative models, seems crucial. Demonstrating strong performance against these more recent models would be beneficial.**
> >
> > First, We attempted to incorporate a comparison with NDP (Dutordoir et al, 2023), but encountered difficulty in replication. Nevertheless, we have included a detailed comparison and analysis with NDP in Appendix A.8. Essentially, NDP doesn’t address the known theoretical flaws of Neural Processes (NPs) and introduces two additional problems: (i) NDP fails to induce a valid stochastic process as it does not satisfy the marginal consistency criterion required by Kolmogorov Extension Theorem (Kolmogorov et al, 2018), and (ii) it relies on uncorrelated Gaussian noise for denoising, which is not applicable in function. Conversely, OFM establishes a more theoretically sound framework by rigorously defining learning within function spaces. Additionally, Bayesian functional regression within the OFM framework adheres to valid stochastic processes, offering a robust and theoretically grounded solution
> >
> > Reference:
> >
> > Andre˘ı Nikolaevich Kolmogorov and Albert T Bharucha-Reid. Foundations of the theory of probability: Second English Edition. Courier Dover Publications, 2018. ISBN 0-486-82159-5
> >
> > Vincent Dutordoir, Alan Saul, Zoubin Ghahramani, and Fergus Simpson. Neural Diffusion Processes, June 2023. URL http://arxiv.org/abs/2206.03992. arXiv:2206.03992 [cs, stat].
> >
> > **Q4. Moreover, recent NP models tackle not only 1D problems but also more complex tasks, such as predicting full RGB-channel images from partially observed 2D domains. This image completion task is challenging, as it lacks the smooth properties typical of PDEs, yet these models show promising performance. I am curious to know if the proposed approach can also handle such complex image functions effectively.**
> >
> > This for suggestions regarding co-domain functional regression problems. In response, we have added an additional experiment in Appendix A.4, demonstrating our framework can be directly applied to co-domain settings (functions with multiple channels).
> > We agree that problems related to pictures and images are challenging, however, we made the focus of this study on a much broader and immediate problem of regression for scientific domains. Importantly, please note that most of the neural operator architectures are designed for scientific domains and are yet to be advanced for the domain of images. There have been a few initial studies on developing neural operators for the operator view of images, but the architectures are still in their preliminary state. Therefore, running such experiments at the moment brings artificial disadvantages to SPL methods. To this end, we have also provided experiments on continuous signed distance function MNIST-SDF examples.

---

> > > ### Author Response · Authors · 2024-11-25
> > >
> > > Dear Reviewer pMpj,
> > >
> > > We sincerely appreciate the time and effort you've dedicated to reviewing our paper and are grateful for your positive feedback on our methods and results.
> > >
> > > As the author-reviewer discussion period is extended, we kindly request you to review our responses and let us know if you have any additional concerns or questions. We are more than happy to address any remaining issues.
> > >
> > > We would also greatly appreciate it if you could consider revisiting your score based on our responses and the feedback from other reviewers.
> > >
> > > Thank you once again for your thoughtful and constructive review!
> > >
> > > Best regards,
> > > Authors

---

### Official Review · Reviewer_MbXF · 2024-11-03

**Soundness:** 3
**Presentation:** 1
**Contribution:** 2
**Rating:** 5
**Confidence:** 3

**Summary:**

This paper proposes a framework for learning stochastic processes parameterized by neural operators by generalizing flow-matching to function spaces. They do so by generalizing the conditional flow-matching objective to function spaces. This enables tractable density estimation on any new collection of points.

**Strengths:**

1. I think the problem of generalizing flow-matching to function spaces is an interesting one.

**Weaknesses:**

## Clarity

1. I found the notation throughout the paper to be extremely unclear. One example of this would be that throughout section 3, $\mathcal{G}$ is defined to be an operator but always applied to a dataset instead of a function. And there are several others throughout the paper. I am okay with some abuse of notation where it helps with readability but in my opinion, this is not the case here. I found it quite hard to keep track of when we are talking about objects in a function space and when we are talking about objects in a domain. I believe the readability can be improved significantly.

2. The authors don't make any effort to disentangle their novel contribution from what is already known in the literature. They have a relatively long background section and ideally, the distinction between the methods section and the background section would have served to make it clear what is new in this paper and what is already known in the literature. But throughout the methods section, they refer to results known in the literature without clarifying whether they are being applied, adapted, or extended, making it hard to identify the paper’s unique contributions.

## Novelty

The objective proposed in this paper is a relatively straightforward generalization of the conditional flow-matching objective. While the extension to function spaces may offer some theoretical appeal, it doesn’t represent a particularly groundbreaking advancement beyond established work as other works have already generalized flow-matching to function space.

## Experiments
Basically, all the details of the experimental setup are missing. There are no details about the FNO used to parameterize the operator, no wall-clock times for training, or the number of Hutchinson samples used from equation 22. Table 1 is also missing error bars.

**Questions:**

See weaknesses. Additionally, why is the following equation in page 3 true:
$$ \mathbb{P}( \{ u(x_1) \dots, u(x_m) \} ) = J\mathcal{G}\Bigr\rvert_{ \{ a(x_1)\dots,a(x_n) \} }\mathbb{P} \{ a(x_1)\dots,a(x_n) \} $$

---

> ### Author Response · Authors · 2024-11-22
> **Official Comment by Authors Part I**
>
> We thank the reviewer for the constructive and insightful feedback,  which definitely helps us enhance the clarity and depth of our manuscript.  We are also delighted that the reviewer found generalizing flow-matching to function spaces interesting.
>
> It is worth emphasizing that our main contribution lies in developing the first simulation-free ODE framework for functional regression purposes, demonstrating superior performance over existing baselines. The theory development for generalizing flow matching to stochastic processes as well as the development of optimal-transport infinite-dimensional flow matching are considered as additional contributions.  We further elaborate on the contributions in the latter part of the introduction.
>
> In the following, we address the comments and explain changes made in the main text. These changes in the main text are colored in blue in the revised draft.
>
> **Q1. I found the notation throughout the paper to be extremely unclear. One example of this would be that throughout section 3,  is defined to be an operator but always applied to a dataset instead of a function. And there are several others throughout the paper. I am okay with some abuse of notation where it helps with readability but in my opinion, this is not the case here. I found it quite hard to keep track of when we are talking about objects in a function space and when we are talking about objects in a domain. I believe the readability can be improved significantly.**
>
> We apologize for any confusion caused by our writing and notation, we have carefully revised the text to clarify these aspects. In section 3, we follow the convention in defining a stochastic process by using a collection of points rather than continuous function (e.g. in GP definition https://en.wikipedia.org/wiki/Gaussian_process#Definition)
>
> The $\mathcal{G}$ is an operator applied on functions, e.g., “a”. This function is often realized at some discretization points, e.g., $\\{x_1,…,x_n\\}$. In section 3, as the reviewer mentioned, we abuse the notation and apply $\mathcal{G}$ on this discretization of “a” on the collection of points/positions $\\{x_1,\cdots,x_n\\}$, i.e., $\\{a(x_1),…,a(x_n)\\}$.
>
> Please note that we don’t refer to $\\{a(x_1),... a(x_n)\\}$ or $\\{u(x_1), .. u(x_n)\\}$ as a dataset. This is a point evaluation of a function. The notation datasets in operator learning works are slightly different. To be specific, the training dataset is the collection of N discretized function $\\{u_i|D_i\\}_i^N$, where $u_i|D_i$ is a discretized observation of the $u_i$ function. In practice, for the convenience of preparing the training dataset, we tend to choose $D_i$ the same for all function samples, e.g.  $D_i := \\{x_1, …, x_n\\}$ regardless of the sample index “i”
>
> Section 3 is provided to build the link between the infinite-dimensional flow matching and its finite-marginal. (The former highly relies on Gaussian measure, while the latter one concerns with the Lebesgue measure). We also add a section in Appendix A.2 to elaborate how to generalize flow matching to stochastic process via Kolmogorov extension theorem, which link the finite-dimensional marginal equipped with Lebesgue measure to the probability measure of a stochastic process in infinite-dimensional space.
>
> **Q2. The authors don't make any effort to disentangle their novel contribution from what is already known in the literature. They have a relatively long background section and ideally, the distinction between the methods section and the background section would have served to make it clear what is new in this paper and what is already known in the literature. But throughout the methods section, they refer to results known in the literature without clarifying whether they are being applied, adapted, or extended, making it hard to identify the paper’s unique contributions.**
>
> This is a very critical point, we apologize for not addressing this issue in our previous draft. We have now added a paragraph in the introduction to clarify our unique contribution. Additionally, in Appendix A.8 of the revised draft, we make detailed comparisons with related models or baselines, highlighting our novel contributions and potential limitations.

---

> > ### Author Response · Authors · 2024-11-22
> > **Official Comment by Authors Part II**
> >
> > **Q3. Basically, all the details of the experimental setup are missing. There are no details about the FNO used to parameterize the operator, no wall-clock times for training, or the number of Hutchinson samples used from equation 22. Table 1 is also missing error bars.**
> >
> > In the updated draft, we have detailed the experimental setup and parameters in Appendix A.7, we also uploaded the code in the supplementary. Regarding Table1, it’s common to omit error bars in functional regression tasks, since the metric used (MSLL, SMSE) compare the goodness-of-fit between two distributions, which naturally incorporates the uncertainty. For example, related studies [Maronas,et al. 2021] & [Salimbeni et al, 2017] do not include error bars for all tables
> >
> > Reference:
> >
> > Juan Maroñas, Oliver Hamelijnck, Jeremias Knoblauch, and Theodoros Damoulas. Transforming Gaussian processes with normalizing flows. pp. 1081–1089. PMLR, 2021. ISBN 2640-3498
> >
> > Hugh Salimbeni and Marc Deisenroth. Doubly stochastic variational inference for deep Gaussian processes. Advances in neural information processing systems, 30, 2017.
> >
> > **Q4. Additionally, why is the following equation in page 3 true:
> > $\mathbb{P} ( \\{ u(x_1), u(x_2), \ldots, u(x_n) \\}) = \textbf{J} \mathcal{G}  \Big|_{\\{a(x_1), a(x_2), \ldots, a(x_n)\\}} \mathbb{P}( \\{ a(x_1), a(x_2), \ldots, a(x_n) \\})$**
> >
> > That is the change of variables for two random vectors defined on $\mathbb{R}^d$, where $\textbf{J} \mathcal{G}  \Big|_{\\{a(x_1), a(x_2), \ldots, a(x_n)\\}}$ is the Jacobian.
> > (https://en.wikipedia.org/wiki/Probability_density_function#Vector_to_vector)

---

> > > ### Comment · Reviewer_MbXF · 2024-11-26
> > >
> > > Thank you for your response.
> > >
> > > "That is the change of variables for two random vectors defined ... is the Jacobian. (https://en.wikipedia.org/wiki/Probability_density_function#Vector_to_vector)"
> > >
> > > I don't understand how the formula on the paper relates to the one in the link which is of course the well-known change of variables formula. The link contains the determinant of the inverse of the Jacobian, unlike the formula in the paper. Also in the paper G is an operator, not a function. I think this is another case where the notation is not clear.
> > >
> > > In general, I concur with the other reviewer. I think this paper requires extensive rewriting to improve the clarity. So I will keep my score.

---

> ### Author Response · Authors · 2024-11-25
>
> Dear Reviewer  MbXF,
>
> We sincerely appreciate the time and effort you've dedicated to reviewing our paper and are grateful for your positive feedback on our methods and results.
>
> As the author-reviewer discussion period is extended, we kindly request you to review our responses and let us know if you have any additional concerns or questions. We are more than happy to address any remaining issues.
>
> We would also greatly appreciate it if you could consider revisiting your score based on our responses and the feedback from other reviewers.
>
> Thank you once again for your thoughtful and constructive review!
>
> Best regards,
> Authors

---

> ### Author Response · Authors · 2024-11-26
>
> That's an excellent point. In operator learning, neural operators are typically designed to map an input function to an output function. When the input function is provided at a specific discretization (e.g., a set of points with their corresponding values), the model processes this discretized input as a collection of points and their values. Traditionally, in operator learning, this process is seen as an approximation of the operator's application to the underlying continuous function, where the discretization introduces approximation errors. Thus, the input is conceptually still treated as a function.
>
> Moreover, the application of the operator to a collection of points is well-defined, and, by the discretization convergence theorem, as the number of points increases, this operation converges to a well-defined mapping.
>
> In this paper, leveraging these properties, we adopt a different perspective as described in the introduction. We extend neural operators to define explicit maps between collections of points. In this framework, the input is not the abstract function itself but rather a collection of points and their associated values. Importantly, this mapping remains well-defined regardless of the number of points in the collection and, by the discretization convergence theorem, converges to a unique mapping as the point collection approaches the underlying continuous function.
>
> With reference to the specified equation, we map values from one collection of points to another under the operator G, which transforms the measure from multivariate Gaussian using the Jacobian. Let's instead look at this Wikipedia link: https://en.wikipedia.org/wiki/Flow-based_generative_model#Derivation_of_log_likelihood. As noted in the Wikipedia link, the det of inverse ($\mathcal{G}$) is equal to the inverse of det of $\mathcal{G}$. Maybe, we can refer to this Wikipedia page. We can use this link to carry on the discussion.
>
> Last, similar transformations are already deployed in relevant works. For instance, in Transforming GP [Maronas,et al. 2021]., the authors employ a marginal normalizing flow, which acts as a point-wise operator to transform values from a GP to another. Consequently, the induced Jacobian is a diagonal matrix. More recently, OpFlow [Shi et al, 2024] introduces an invertible neural operator by generalizing RealNVP to function space, which induces a triangular Jacobian matrix. In our work, we extend this framework to a more comprehensive case: a diffeomorphism. Here, the induced Jacobian is a full-rank matrix and is not necessarily triangular or diagonal.
>
> Reference:
>
> Juan Maroñas, Oliver Hamelijnck, Jeremias Knoblauch, and Theodoros Damoulas. Transforming Gaussian processes with normalizing flows. pp. 1081–1089. PMLR, 2021. ISBN 2640-3498
>
> Yaozhong Shi, Angela F. Gao, Zachary E. Ross, and Kamyar Azizzadenesheli. Universal Functional
> Regression with Neural Operator Flows, April 2024  https://arxiv.org/abs/2404.02986

---

### Official Review · Reviewer_mZQU · 2024-11-03

**Soundness:** 3
**Presentation:** 1
**Contribution:** 1
**Rating:** 3
**Confidence:** 3

**Summary:**

Using neural operators, this paper extends flow matching based continuous normalizing flows to infinite dimensions, to learn stochastic processes. By virtue of finite-dimensional densities being tractable, these stochastic processes can be used as prior distributions for subsequent Bayesian inference, which authors propose doing using stochastic gradient Langevin dynamics.

The approach is empirically tested on synthetic and more realistic datasets, showing improvements.

**Strengths:**

- I think that the overall idea of using infinite-dimensional flow matching for learning stochastic process priors is a good one.
- The empirical advantage over GP baselines is convincing in the black hole and MNIST setups.
- Whenever authors use some ideas, they mention the sources of those ideas (provide citations).

**Weaknesses:**

The paper brings together numerous components from modern literature to create something new. In my opinion, it's not hard to imagine that something along the lines of this paper would be possible. Working out the details, again in my view, doesn't require a lot of new ideas but might demand significant effort. I believe that papers of this nature can be quite useful and make worthy publications. However, they should (1) provide high-quality code, so future researchers don't have to replicate the work themselves, and (2) clearly present the "diff": the differences compared to existing methods and tools, highlighting the details that proved challenging, and aspects potential users should be aware of. Unfortunately, I think this paper falls short in these areas:
- Writing is sloppy (a far-from-exhausting list of examples is given below in the “Questions” section, as evidence). It feels like nobody even tried to proofread the text before submission.
- There is no clear “diff”, instead, the text concentrates on reviewing the prior work or something that only marginally differs from the prior work.
- There is no code in the supplementary.

I don’t think that the method is unworthy of publication. However, I think that the paper, in today's form, is. The text should be majorly improved, the code should be provided.

**Questions:**

Questions/major comments (some are probably consequences of sloppy writing):
- In Figure 1, it looks like a Gaussian process is just chosen poorly. First of all, it does not look smooth enough (=> e.g., use RBF instead of Matérn, use a larger length scale). Second, since here you obviously have a diagonal trend, you could consider a matrix ARD: when instead of one length scale parameter $l$ that transforms $x \to x/l$, you learn a matrix $A$ that transforms $x \to A x$.
- Line 091, you say that GP parameters in deep GPs are “manually tuned”. This is plainly not true, they are typically optimized during the variational inference procedure.
- Lines 092-095. “Warped GPs (Kou et al., 2013) and transforming GP (Maroñas et al., 2021) methods use historical data to learn a pointwise transformation of GP values and outperform deep GP type methods).” This is a strong and controversial claim without any evidence to support it.
- Lines 096-097, I don’t think neural processes are a variational inference approach, this is certainty not how they were originally introduced.
- You are not careful when going into the infinite-dimensional setting, and going infinite-dimensional is perhaps the most important contribution of the paper. For example, in Equation (11) you use the gradient, but you don’t talk about the precise meaning of this operation in infinite dimension.

Examples of sloppy writing:
- The content goes over the right margin on multiple occasions: in the display-style formula at the bottom of page 3, in Equation (24), in Table 1.
- Figures contain barely readable font sizes.
- Lines 042-044, the clause “allowing us to learn probability priors over the stochastic process, ergo, sampling of any collection points with their associated density.” You are not learning a prior over stochastic process, you are learning a prior over functions, which _is_ a stochastic process. Also, the word “sampling” typically means “random sampling” throughout the paper, I don’t think here you mean “random sampling”.
- “Other works in this area have following the success of flow matching and diffusion models”. I guess it was supposed to be “followed”, the typo makes this sentence very hard to read.
- In the second paragraph you talk about the “universal functional regression” of Shi et al. that makes it feel like a classical approach, whereas it is a paper from 2024. Also you vaguely overload the standard term “functional analysis”.
- Line 115. The notation for a probability space and a measurable space is introduced, which I believe is never subsequently used.
- Throughout the text: it is very strange that you use the curly braces {} for vectors, instead of the standard () or somewhat less standard [], especially since curly braces are standard for something else, for sets.
- Lines 151-154 The sentence “More specifically, for m ≥ n points at which the function is to be inferred, … display-style formula …” does not make sense, grammatically.
- At the time Figure 1 is first referred to, “OFM” is an undefined abbreviation.
- Line 049, “deep GP” -> “deep GPs”.

The list could go on and on.

---

> ### Author Response · Authors · 2024-11-22
> **Official Comment by Authors Part I**
>
> We appreciate the reviewer’s valuable comments and the positive remarks regarding the innovative approach of using infinite-dimensional flow matching for learning stochastic process priors and our empirical results. We acknowledge the concerns you have raised regarding writing and code, and are committed to addressing them comprehensively. In the following, we address the comments and explain changes made in the main text. These changes in the main text are colored in blue in the revised draft.
>
> **Q1. Writing is sloppy (a far-from-exhausting list of examples is given below in the “Questions” section, as evidence). It feels like nobody even tried to proofread the text before submission.**
>
> We apologize for any confusion caused by our writing and sincerely appreciate your attention to detail. We are committed to a thorough revision of the writing and all other details.
>
> **Q2. There is no clear “diff”, instead, the text concentrates on reviewing the prior work or something that only marginally differs from the prior work.**
>
> This is a great comment, we added a paragraph in the introduction to further summarize our contributions. Additionally, we make a detailed comparison with related models or existing baselines in Appendix A.8 of the revised draft, highlighting our unique contributions and potential limitations
>
> **Q3. There is no code in the supplementary.**
>
> We apologize for not including the code initially. Now, we have included the code in the supplementary with all experiments details. We will release easy-to-use code with tutorials after publication.
>
> **Q4. In Figure 1, it looks like a Gaussian process is just chosen poorly. First of all, it does not look smooth enough (=> e.g., use RBF instead of Matérn, use a larger length scale). Second, since here you obviously have a diagonal trend, you could consider a matrix ARD…**
>
> This is an insightful observation. We now include two scenarios for the regression with Navier-Stokes functional data in Figure 6 and 1, with and without the diagonal trend. We have chosen the RBF kernel for the GP, whose parameters are optimized via Maximum Likelihood Estimation (maximizing the log-marginal likelihood). However, the conclusions remain the same.
>
> **Q5. Line 091, you say that GP parameters in deep GPs are “manually tuned”. This is plainly not true, they are typically optimized during the variational inference procedure.**
>
> We apologize again for our writing and address this in the main text, the revised text reads “The parameters of deep GPs are commonly optimized by minimizing the variational free energy, which serves as a bound on the negative log marginal likelihood”
>
> **Q6.  Lines 092-095. “Warped GPs (Kou et al., 2013) and transforming GP (Maroñas et al., 2021) methods use historical data to learn a pointwise transformation of GP values and outperform deep GP type methods).” This is a strong and controversial claim without any evidence to support it.**
>
> We have changed the text to “Warped GPs (Kou et al., 2013) and transforming GP (Maroñas et al., 2021) methods use historical data to learn a pointwise transformation of GP values and achieve on par performance compared to deep GP type methods.”
>
> **Q7. Lines 096-097, I don’t think neural processes are a variational inference approach, this is certainty not how they were originally introduced.**
>
> We have revised the text to read “ inspired by variational inference”.
>
> **Q8. You are not careful when going into the infinite-dimensional setting, and going infinite-dimensional is perhaps the most important contribution of the paper. For example, in Equation (11) you use the gradient, but you don’t talk about the precise meaning of this operation in infinite dimension.**
>
> Equation (11) represents the transport equation or continuity equation that exists in function space, as seen in the first equation of Chapter 8 (Ambrosio et al., 2008). This equation, which expresses the conservation of mass in infinite-dimensional spaces, is widely applicable in fields such as fluid dynamics and electromagnetism. The symbol "$\nabla \cdot$" denotes the divergence operation
>
>
> Reference:
>
> Luigi Ambrosio, Nicola Gigli, and Giuseppe Savaré. Gradient flows: in metric spaces and in the space of probability measures. Springer Science & Business Media, 2008. ISBN 3-7643-8722-X

---

> > ### Author Response · Authors · 2024-11-22
> > **Official Comment by Authors Part II**
> >
> > **Q9. The content goes over the right margin on multiple occasions: in the display-style formula at the bottom of page 3, in Equation (24), in Table 1.**
> >
> > Thank you for your suggestions, we have addressed these formatting issues
> >
> > **Q10. Figures contain barely readable font sizes.**
> >
> > We have adjusted font sizes for all figures to ensure clarity
> >
> > **Q11. Lines 042-044, the clause “allowing us to learn probability priors over the stochastic process, ergo, sampling of any collection points with their associated density.” You are not learning a prior over stochastic process, you are learning a prior over functions, which is a stochastic process. Also, the word “sampling” typically means “random sampling” throughout the paper, I don’t think here you mean “random sampling”.**
> >
> > Sorry for the confusion caused. It was incorrectly worded. We have changed the text to “ sampling the values of any collection points with their associated density“. The points themselves are indeed not sampled, they are picked. Only their values are sampled. Thank you for pointing this out.
> >
> > **Q12. “Other works in this area have following the success of flow matching and diffusion models”. I guess it was supposed to be “followed”, the typo makes this sentence very hard to read.**
> >
> > This is a typo, we have fixed it in the main text.
> >
> > **Q13.  In the second paragraph you talk about the “universal functional regression” of Shi et al. that makes it feel like a classical approach, whereas it is a paper from 2024. Also you vaguely overload the standard term “functional analysis”.**
> >
> > We have addressed this in the main text, we changed the text to “Learning the prior over the process is crucial for universal functional regression, which is a recently proposed Bayesian method for functional regression”
> >
> > **Q14. Line 115. The notation for a probability space and a measurable space is introduced, which I believe is never subsequently used.**
> >
> > We adhere to the convention of defining a stochastic process, where the concepts for probability space and measure space are used. While not frequently used, we believe defining the spaces make the set up more concrete.
> >
> > **Q15.  Throughout the text: it is very strange that you use the curly braces {} for vectors, instead of the standard () or somewhat less standard [], especially since curly braces are standard for something else, for sets**
> >
> > Thank you for pointing this out. When defining stochastic processes, often vector notation is used, which requires defined vector spaces for any collection of points where the order matters. At the same time, often, for convenience, collection/set of values convection is used (some textbooks use both) which sometimes make the notation less heavy and implies only the joint distribution matters. In this paper, we adopted this convention. As the reviewer noted,  $\\{a(x_1), … a(x_n)\\}$ is not a dataset but a collection of point evaluations of an underlying function. We elaborate this point in Appendix A.2.
> >
> > **Q16. Lines 151-154 The sentence “More specifically, for m ≥ n points at which the function is to be inferred, … display-style formula …” does not make sense, grammatically.**
> >
> > We fixed this in the main text.
> >
> > **Q17. At the time Figure 1 is first referred to, “OFM” is an undefined abbreviation**
> >
> > We have addressed this in the revised draft by using the full name “Operator Flow Matching (OFM)”
> >
> > **Q18.  Line 049, “deep GP” -> “deep GPs”.**
> >
> > We have addressed this in the revised draft.

---

> > > ### Comment · Reviewer_mZQU · 2024-11-22
> > >
> > > I acknowledge reading the authors' rebuttal. While I find the methodological content of the paper compelling as mentioned in my initial feedback, the original version's writing and presentation quality were significantly below standard. This necessitates a major revision of the paper and its comprehensive re-review, which is beyond the scope of the rebuttal stage. Consequently, I stand by my recommendation to reject the paper. However, I highly encourage the authors to revise and resubmit the paper.

---

> > > > ### Author Response · Authors · 2024-11-25
> > > >
> > > > Dear Reviewer mZQU,
> > > >
> > > > We sincerely appreciate the time and effort you've dedicated to reviewing our paper and are grateful for your positive feedback on our methods and results.
> > > >
> > > > As the author-reviewer discussion period is extended, we kindly request you to review our updated draft and let us know if you have any additional concerns or questions. We are more than happy to address any remaining issues.
> > > >
> > > > We would also greatly appreciate it if you could consider revisiting your score based on our responses and the feedback from other reviewers.
> > > >
> > > > Thank you once again for your thoughtful and constructive review!
> > > >
> > > > Best regards,
> > > > Authors

---

### Author Response · Authors · 2024-11-22
**Changes made in the revised draft**

We would like to thank all reviewers for their insightful and valuable feedback, which has significantly helped us enhance the clarity and quality of our work. Below, we outline the key modifications made during the first revision. Changes in the main text are highlighted in blue in the revised draft for easier identification.

- Carefully revised the text, adjusted the font size for all figures, and uploaded the code to the supplementary materials. Additionally, we added a paragraph in the introduction to further clarify our contribution.

-  Added detailed analysis and comparison with related models or baselines in Appendix A.8, highlighting our contribution and potential limitations

-  Provided regression results on finer grids (currently 64x64, previous 32x32) for Navier-Stokes, Black hole, MNIST-SDF examples.
-  Added detailed experimental setup and parameters used in Appendix A.7
-  Added the SGLD algorithm used in this study in Appendix A.5
-  Added one co-domain regression experiment in Appendix A.4
-  Added more posterior sampling results in Appendix A.3
-  Added the background and proof for generalizing flow matching to stochastic process with Kolmogorov extension theorem, provided in Appendix A.2

---

### Meta-Review · Area_Chair_HRLB · 2024-12-20

**Metareview:**

This paper proposes a flow matching method in infinite-dimensional function spaces. The proposed Operator Flow Matching (OFM) framework extends standard flow matching methods to infinite-dimensional spaces by constructing a framework for learning functional vector fields. The suggested stochastic process enables the computation of finite-dimensional densities, making it applicable as a prior distribution for Bayesian inference. The authors propose inference using Stochastic Gradient Langevin Dynamics (SGLD). While the paper presents an important and highly intriguing approach, many reviewers pointed out mathematical issues and presentation problems. Due to numerous typographical errors and unclear points in the mathematical formulations, substantial revisions are necessary before the paper can be considered for publication. I strongly encourage the authors to refine these aspects and resubmit the paper.

**Additional Comments On Reviewer Discussion:**

Reviewer mZQU, Reviewer MbXF, and Reviewer pMpj highlighted issues with the presentation quality, including unclear mathematical notations and other ambiguities. While the authors attempted to address these concerns during the discussion phase, the reviewers still indicated that significant revisions are necessary.

---

### Decision · Program_Chairs · 2025-01-22

Reject